# Understanding Cancer’s Defense against Topoisomerase-Active Drugs: A Comprehensive Review

**DOI:** 10.3390/cancers16040680

**Published:** 2024-02-06

**Authors:** Nilesh Kumar Sharma, Anjali Bahot, Gopinath Sekar, Mahima Bansode, Kratika Khunteta, Priyanka Vijay Sonar, Ameya Hebale, Vaishnavi Salokhe, Birandra Kumar Sinha

**Affiliations:** 1Cancer and Translational Research Centre Dr. D.Y. Patil Biotechnology & Bioinformatics Institute, Dr. D.Y. Patil Vidyapeeth, Pune 411033, Maharashtra, India; nilesh.sharma@dpu.edu.in (N.K.S.); anjali.bahot@dpu.edu.in (A.B.); gopinath254867@gmail.com (G.S.); mahima.bansode@dpu.edu.in (M.B.); 20050131@dpu.edu.in (K.K.); 21050181@dpu.edu.in (P.V.S.); 20050017@dpu.edu.in (A.H.); vaishnavisalokhe1991@gmail.com (V.S.); 2Mechanistic Toxicology Branch, Division of Translational Toxicology, National Institute of Environmental Health Sciences, Research Triangle Park, Durham, NC 27709, USA

**Keywords:** topoisomerases, drug resistance, molecular heterogeneity, neoplasms, combinatorial approaches

## Abstract

**Simple Summary:**

Increasing number of cancer patients and significant mortality among these patients are attributed to many factors including anticancer drug resistance. Among a pool of available anticancer drugs, topoisomerase-active drugs, e.g., doxorubicin and topotecan are used by cancer patients at high dose and that leads to various side effects and post anticancer therapies complications. Therefore, efforts are warranted to understand the reasons behind such anticancer drug resistance so that better therapeutic management of cancer patients can be achieved. Such efforts could be achieved in the preclinical and clinical laboratory settings so that safer and highly efficacious anticancer drug combinations could be explored for the cancer patients. In the future, combinatorial anticancer drug approaches targeting topoisomerase enzymes can combine newer tools and technologies aided by artificial intelligence and machine learning for rapid and reliable validations of anticancer effects by combinatorial therapeutic approaches.

**Abstract:**

In recent years, the emergence of cancer drug resistance has been one of the crucial tumor hallmarks that are supported by the level of genetic heterogeneity and complexities at cellular levels. Oxidative stress, immune evasion, metabolic reprogramming, overexpression of ABC transporters, and stemness are among the several key contributing molecular and cellular response mechanisms. Topo-active drugs, e.g., doxorubicin and topotecan, are clinically active and are utilized extensively against a wide variety of human tumors and often result in the development of resistance and failure to therapy. Thus, there is an urgent need for an incremental and comprehensive understanding of mechanisms of cancer drug resistance specifically in the context of topo-active drugs. This review delves into the intricate mechanistic aspects of these intracellular and extracellular topo-active drug resistance mechanisms and explores the use of potential combinatorial approaches by utilizing various topo-active drugs and inhibitors of pathways involved in drug resistance. We believe that this review will help guide basic scientists, pre-clinicians, clinicians, and policymakers toward holistic and interdisciplinary strategies that transcend resistance, renewing optimism in the ongoing battle against cancer.

## 1. Introduction

Demographic projections suggest that by 2040 there will be 16.3 million cancer deaths worldwide, with 27.5 million new instances of the disease. Data also show that cancer has caused 10 million deaths globally, with breast, lung, colon, rectal, prostate, skin, and stomach cancer being the most common types [1,2]. Cancers are highly heterogeneous with many subtypes and molecular profiles, and thus present complex challenges. Tumor heterogeneity at cellular levels is related to genetic and environmental factors. Lifestyle choices, modes of preventive strategies, and modalities of therapies for cancer patients also play roles in inducing tumor heterogeneity, leading to chemotherapy resistance to various anticancer drugs including topoisomerase-active drugs (topo-active drugs) [3,4,5].

Chemotherapy resistance, a formidable obstacle in cancer treatment, arises from gene mutations, gene amplification, or epigenetic changes affecting the uptake, metabolism, or export of drugs from cells. Acquired resistance further involves intricate interactions with intracellular and extracellular signaling pathways, as well as metabolic adaptations [6,7]. Due to these extreme challenges in therapy, both the understanding of the mechanisms of topo-active drug resistance and innovative approaches to successful therapy are urgently needed.

Topoisomerase enzymes are crucial enzymes involved in DNA replication, chromosomal segregation, transcription, and recombination. They exist in two classes: Topoisomerase I (TOPI), which cleaves one DNA strand, and Topoisomerase II (TOPII), which cuts simultaneously on both strands [8,9]. Camptothecin (CPT), an alkaloid, discovered in 1958, is an inhibitor of TOPI. FDA-approved derivatives like topotecan and irinotecan target TOPI, while TOPII inhibitors like the anthracycline-based drug doxorubicin (DOX) cause cytotoxic DNA double-strand breaks [10,11,12]. Although these topo-active drugs are extremely effective and utilized to treat a wide variety of cancers, they face the common hurdle of resistance development in tumor cells, necessitating novel approaches.

A prominent contributor to multidrug resistance is the ABC transporter (ATP-binding cassette) protein family, with P-glycoprotein (P-gp)/ATP-binding cassette subfamily B member 1 (ABCB1) being the most studied. Overexpression of the ABC transporter contributes to the efflux of topo-active drugs and is a recognized mechanism [13,14,15,16]. Hence, combinatorial approaches that can target both ABC transporters and TOPI and TOPII enzymes concomitantly may lead to better drug responsiveness.

Glutathione (GSH) is one of the most abundant low-molecular-weight non-protein thiols, immensely regulates the physiological levels of reactive oxygen species (ROS), and, in turn, mediates oxidative stress response during genotoxic drug responses [17,18,19,20,21]. The modulation of GSH-mediated oxidative stress in cancer cells is suggested as an important step to mitigate drug resistance and achieve better drug responsiveness. 

Genotoxic chemicals, including topo-active drugs, induce enhanced DNA repair in cancer cells, allowing them to survive and proliferate. Cancer cells develop resistance mechanisms by modulating DNA repair response pathways, enhancing their survival during genotoxic insults [22,23,24,25].

In immunotherapy-based resistance, the primary resistance to checkpoint inhibitors occurs due to both intrinsic and extrinsic factors, preventing immune response and antigen recognition [26,27,28]. However, combining immune checkpoint inhibitors with topo-active drugs is recommended to minimize resistance.

Cancer stem cells (CSCs) play a crucial role in the formation, growth, and resilience of tumors [29,30,31,32]. Understanding CSC pathways provides insight into combined approaches for effective anticancer drug treatments [33,34]. 

Increased lipid metabolism, glutaminolysis, glycolysis, mitochondrial biogenesis, and the pentose phosphate pathway are all aspects of metabolic reprogramming in cancer that play a significant role in survival and resistance against topoisomerase-active drugs [35,36]. For instance, the Warburg effect and metabolic reprogramming are crucial events during the survival and resistance strategies of cancer cells against topo-active drugs [37,38,39,40]. Therefore, a holistic understanding at the molecular, preclinical, and clinical levels is crucial for developing innovative approaches, such as combining metabolite mimetics with topoisomerase-active drugs, to enhance responsiveness and induce cancer cell death.

Based upon the lacunae in current understanding of topo-active drug resistance in cancer cells, this review offers insights into diverse adaptations at the molecular and cellular levels. These adaptations encompass but are not limited to ABC transporters, oxidative stress, DNA damage response mechanisms, immune heterogeneity, plasticity of CSCs, and metabolic reprogramming. Moreover, we extend prospective avenues for synergistic pharmacological strategies by combining small-molecule inhibitors with topo-active agents. The overarching goal is to enhance the efficacy of cancer cell death while concurrently ameliorating the adverse effects attributed to topo-active drugs.

## 2. Topoisomerases

DNA topoisomerases play a pivotal role in fundamental cellular processes such as DNA replication, transcription, recombination, and chromatin remodeling [41,42]. These enzymes exert control over the topological status of the DNA double helix by inducing either single-strand (TOPI) or double-strand (TOPII) DNA breaks [41,42]. 

TOPI, a 100 kDa monomeric protein, is expressed by a single copy gene on chromosome 20q12-13.2, which must be phosphorylated for its catalytic activity [41]. The TOPI cleaves one strand of DNA [41]. It is not an ATP-dependent enzyme (reverse gyrase is an exception). TOPI can also be further divided into type IA (TOP3α and TOP3β) and type IB (TOPI and human mitochondrial topoisomerase) enzymes based on whether the protein is linked at a 5′-phosphate or 3′-phosphate [41,43]. The functional role of TOPI is paramount, as it is indispensable for various cellular processes, contributing to the synchronous dynamic of DNA structure and function.

Parallelly, TOPII emerges as a critical player, essential for chromosome segregation and reprogramming replicons. Topo II inhibition impairs the entirety of DNA replication, leading to the stabilization of replication protein A (RPA) onto (ssDNA) [42,44]. This intricate connection illustrates the importance of TOPII in maintaining genomic stability and integrity [42,44].

The significance of topoisomerases in cellular processes has encouraged extensive research efforts, particularly in the context of pharmacological intervention. Various agents, including irinotecan, topotecan, doxorubicin (DOX), and etoposide (VP-16), have been developed to target these enzymes [45,46,47]. The focus on these pharmacological agents emphasized the potential for therapeutic actions aimed at modulating topoisomerase activity, paving the way for critical cellular processes and, consequently, diseases associated with DNA structural dynamics.

## 3. Mechanisms of Action of Topo-Active Drugs

TOPI and TOPII are vital cellular targets of many chemotherapeutic drugs, playing a pivotal role in orchestrating the intricate structure of chromatin, a dynamic amalgamation of DNA and proteins within cells. Chemotherapeutic agents that inhibit TOPI and TOPII induce distinctive effects on chromatin [45,46,47,48].

TOPI, a preferred target for many chemotherapeutics, serves as a crucial indicator of malignant cell proliferation [46,47,48,49,50,51,52]. TOPI inhibitors, categorized as TOPI poisons and TOPI suppressors, act at the TOPI-DNA complex level, promoting DNA breakage [51]. TOPI poisons like topotecan induce single-stranded breaks after DNA cleavage, inhibiting ligation. The overexpression of TOPI is correlated with an increase in the susceptibility of tumor cells to TOPI inhibitors. In tumor cells with minimal TOPI expression, TOPI suppressor activity is higher. As a result, the choice between these inhibitors depends on TOPI expression levels in tumor cells, offering distinct anticancer therapeutic strategies [51].

CPT, a compound altering the topographic state of duplex DNA through single-strand breaks and relegation, targets TOPI [46,47]. CPT stabilizes the covalently bonded TOPI-DNA complex, inhibiting the reconnecting stage of the cleavage/relegation reaction [46,47]. Nemorubicin (MMDX), a third-generation anthracycline derivative, utilizes intercalation in DNA as a TOPI inhibitor, demonstrating efficacy in various tumor models without cardiotoxicity [53,54,55,56,57,58].

Topoisomerase poisons, CPT, and related compounds are key drugs for treating solid tumors. The unique semisynthetic CPT derivative, containing a bulky piperidino side chain at the 10 position, is known as irinotecan. This side chain can be cleaved enzymatically by the enzyme carboxylesterase to 7-ethyl-10-hydroxy-campthothecin (SN38), which is a potent TOPI inhibitor [10,40,51,52]. Topotecan and SN38 both stabilize TOPI cleavage complexes (TOPIcc), which are then converted into DNA damage during DNA replication and transcription [50,51], inhibiting DNA strands from re-ligation, causing permanent replication fork arrest and cell death [24].

TOPII inhibitors exhibit clinical effectiveness against several human malignancies [53,55]. They are classified as TOPII poisons and catalytic inhibitors. DOX and VP-16, popular chemotherapeutic drugs, form a ternary complex with DNA and TOPII, inhibiting transcription and replication and inducing apoptosis in cancer cells [53,54,55,56,57,58,59]. 

Nitric oxide (NO), a physiological signaling molecule, is involved in a variety of cellular activities, including cell growth, survival, and death. It plays a substantial role in detoxifying VP-16 and affects critical cellular proteins, including topoisomerases, but does not affect Adriamycin resistance [60,61]. 

Studies suggest that secondary metabolites produced by plants like alkaloids, terpenoids, polyphenols, and quinones could serve as alternatives to synthetic topoisomerase inhibitors, overcoming drug resistance and other epigenetic changes [62,63]. Terpenoids, a diverse class of plant natural products that include taxol, a complex polyoxygenated diterpenoid, were isolated from the Pacific yew *Taxus brevifolia*, used in treating various cancers. Taxotere, a semisynthetic derivative, enhances water solubility and microtubule polymerization, leading to apoptosis [63]. 

{2-(1-Ethyl-7-methyl-4-oxo-1,4-dihydro-1,8-naphthyridine-3-carbonyl)-N-(m-tolyl)-hydrazinecarbothioamide}, a derivative of nalidixic acid, inhibits both TOPIIα and TOPIIβ, inducing cell cycle arrest at the G2 m phase and apoptosis [64]. Ellipticine, an alkaloid derived from *Ochrosia elliptica labil*, is approved for metastatic breast cancer treatment [65,66]. Additionally, carbazole derivatives also exhibit inhibitory activity against both TOPI and TOPII [67,68,69]. 

## 4. Tumor Heterogeneity and Topo-Active Drugs

Chemotherapy resistance, a pervasive challenge in cancer treatment, particularly against topo-active drugs, represents a multifaceted phenomenon intricately tied to the extensive heterogeneity observed within tumors at various molecular and cellular levels [70,71]. Tumor heterogeneity manifests both intra- and inter-tumorally, influenced by factors such as genome doubling, mutational burden, and somatic copy number alterations. Intra-tumoral heterogeneity, encompassing genetic, epigenetic, neo-antigenic, metabolic, and tumor microenvironment (TME) variations, is a complex landscape [72].

The acidic microenvironment prevalent in tumors creates several obstacles to chemotherapy efficacy, including drug protonation, diminished cellular uptake, elevated MDR1 and P-gp gene expression, heightened MDR proteins, angiogenesis, metastasis, hypoxic conditions, gene downregulation, and diminished chemotherapeutic efficacy [73,74,75,76].

Studies emphasize the critical function of components of the tumor microenvironment (TME), such as cancer-associated fibroblasts (CAFs) and immune suppression cells, in shaping tumor progression. CAFs, through secretion of transformative factors like TGF-beta, HGF, interleukins, and metalloproteinases, modulate the extracellular matrix (ECM) during tumor growth, fostering an environment conducive to chemoresistance [77,78]. Notably, the resistance exhibited by cancer cells, particularly against topo-active drugs, is substantiated by the supportive influence of TME-secreted factors, including those derived from CAFs and immune suppression cells [79,80,81,82,83].

Genetic tumor heterogeneity, exemplified by chromosomal instability, gives rise to cancer genome alterations such as aneuploidy. Aberrant chromosome numbers impact tumor suppressor genes, instigating drug resistance, particularly against topoisomerase-targeting agents [84,85,86,87]. Macro-evolutionary events, signified by substantial chromosomal alterations, contribute significantly to the development of drug resistance to topo-active agents [71,88].

Tumor cells exhibit remarkable epigenetic and phenotypic plasticity, potentially resulting in drug-tolerant persistence and resistance through stable non-genetic alterations in gene expression [89,90,91]. Global epigenetic changes, exemplified by CpG island hypomethylation, augment cell-to-cell variability in cancer cells due to genetic and microenvironmental heterogeneity [87,92,93]. The DNA methylome, a critical epigenetic facet in human cancers, involves promoter CpG island DNA hypermethylation of tumor suppressor genes and global DNA hypomethylation. Cytosine methylation impacts tumorigenicity, as 5-methylcytosine is mutagenic and can undergo spontaneous hydrolytic deamination, resulting in C → T transitions [94,95,96,97]. Clinical and integrated multi-omics analyses suggested the relevance of topoisomerases in tumor heterogeneity [98,99].

## 5. Mutation of Target Enzymes

Resistance to chemotherapeutic drugs can arise due to mutations in genes encoding topoisomerases. In the context of SN38-resistant HCT116 clones, novel mutations were pinpointed in the core subdomain III (p.R621H and p.L617I) and the linker domain (p.E710G), strategically positioned at the interface between these domains. These mutations, while not influencing topoisomerase I (TOPI) expression or activity, induced a reduction in TOPI-DNA cleavage complexes and the formation of double-stranded breaks [100]. 

Concurrently, Adriamycin-selected resistant ovarian cancer (OVCAR-8) cells exhibited heightened resistance to VP-16, exhibiting diminished DNA strand breaks. Mechanistically, resistance encompasses multifaceted factors, such as diminished drug uptake, altered topoisomerase sensitivity, and reduced phosphorylation, collectively attenuating the binding affinity of TOPI and topoisomerase II (TOPII) inhibitors [101,102].

Humans possess two distinct TOPII genes, hTOPIIA on chromosome 17 and hTOPIIB on chromosome 3 [103], each yielding proteins with unique yet overlapping functions. While hTOPIIα is indispensable for cell viability in proliferating cells, hTOPIIβ is expressed in quiescent cells, regulating transcription [104,105,106]. Somatic mutations in TOPII, specifically the ID_TOPIIα pattern, manifest in a mutator phenotype, associated with genomic rearrangements and potential oncogenes in established driver genes. Another mutation type renders the enzyme insensitive to inhibitors, diminishing DNA cleavage activity and elevating drug resistance frequency. These mutations are common but limited due to homozygous requirements [106,107].

Numerous single-nucleotide polymorphisms (SNPs) are evident in both TOPI and TOPII coding genes. Distinct point mutations in the TOPI gene confer resistance to camptothecin (CPT) derivatives. For instance, the substitution of Phe-361 or Asn-722 by Ser induces insensitivity to topotecan. Similarly, a mutation of Ala-653 to Pro in the linker domain imparts resistance to CPT-related drugs [14].

Mutations in TOPII genes are prevalent in resistant cell lines, occurring predominantly in the nucleotide-binding and tyrosine-comprised domains involved in the covalent binding of TOPII to nicked DNA. Notable examples include the Arg 486 to Lys substitution conferring resistance to amsacrine in human leukemia (HL-60) cells and the deletion of Ala 429 resulting in resistance against VP-16 in human melanoma (FEM) cells. Validation of these mutations’ role in drug resistance was accomplished using the yeast expression model [108,109,110].

While research on TOPI and TOPII mutants provides valuable insights, a certain degree of discordance exists regarding their functional relevance. To address this, more study is required elucidating the genetic-based mechanisms of drug resistance in cancer cells for a comprehensive understanding and effective circumvention of resistance [111,112].

## 6. Altered Drug Metabolisms and Topo-Active Drugs

The deregulated transportation of drugs is identified as a potential factor impacting drug metabolic pathways, leading to the inactivation of topo-active drugs. This phenomenon, as outlined in existing research [63,113,114,115,116], holds implications for decreased binding and the formation of inactive metabolites, consequently inducing resistance to topo-active drugs.

The metabolic conversion of DOX results in the production of alcohol metabolites, such as doxorubicinol (DOXol), alongside DOX deoxy aglycone and DOXol hydroxy aglycone. These transformations contribute to altered toxicity and diminished antineoplastic activity [113,115]. Conversely, the reductive metabolism of DOX, facilitated by NADPH (nicotinamide adenine dinucleotide phosphate), yields reactive species with heightened cytotoxicity [117,118]. Notably, the polymorphism of the cytochrome P450 1B1 gene is implicated in reduced sensitivity to various DNA-interacting anticancer agents, including alkylators, CPT, and TOPII inhibitors [119]. Furthermore, an intriguing study supports the notion that nitrogen oxide-derived species can detoxify VP-16 through direct nitrogen oxide radical attack, potentially contributing to increased drug resistance in VP-16-treated cancer patients [63].

The study by Pathania et al. indicates the significant roles played by phase II drug metabolism enzymes, including uridine diphospho-glucuronosyltransferase (UGTs), glutathione-S-transferases (GSTs), dihydropyrimidine dehydrogenase (DPDs), and thiopurine methyltransferases (TPMTs), in inducing resistance to topo-active drugs [120]. Genetic variants of phase I and II enzymes, such as cytochrome P450 (CYP) 3A4 and uridine diphosphate glucuronosyltransferase (UGT) 1A, are associated with the metabolism of irinotecan and VP-16. These genetic variations potentially influence the efficacy and toxicity outcomes [121,122].

Therefore, the efficacies of topo-active drugs and other anticancer agents are intricately linked to the generation of intracellular metabolized products. These products, arising from both non-enzymatic and enzymatic biotransformation, modulate the therapeutic and toxic properties of topo-active drugs. Additionally, the involvement of glutathione (GSH) in the biotransformation of topo-active drugs further contributes to the complex landscape of anticancer drug responses [116].

## 7. ABC Transporter-Mediated Resistance of Topo Drugs

Multidrug resistance (MDR) poses a formidable challenge in chemotherapy, compromising its efficacy and elevating the risk of patient mortality. Extensive research has revealed the intricacies of drug transport regulation, a process tightly governed by members of the ABC transporter protein family. Notably, P-glycoprotein (P-gp/ABCB1), the pioneering ABC transporter, serves as a focal point in this intricate network [13,123,124]. This protein is accountable for an energy-dependent expulsion of intracellular drugs from cancer cells, precipitating diminished drug concentrations and engendering drug resistance, particularly in the context of multidrug resistance (MDR).

Further complicating this landscape, certain efflux proteins, such as P-gp (ABCB1), multidrug proteins (MRPs or ABCCs), and breast cancer resistance protein (BCRP or ABCG2), are localized on the luminal membrane. They are responsible for extruding substrates from blood–brain barrier (BBB) endothelial cells back into the bloodstream, thereby diminishing the central bioavailability of numerous drugs [125]. Within the expansive ABC transporter superfamily, membrane proteins facilitate the extrusion of a diverse array of substrates across cellular membranes. The classification into seven subfamilies, based on sequence similarity and structural organization, underscores the complexity inherent in this molecular repertoire [13,126].

Topo-active drugs, including DOX, VP-16, CPT, and their derivatives, emerge as substrates for ABC transporters, contributing to drug resistance in tumors. The manifestation of ATP-binding cassette transporters in cancer patients with resistant disease further stresses the critical role of ABC transporters in thwarting targeted chemotherapy efforts.

Elucidating the inner workings of ABC transporters during substrate translocation necessitates a comprehensive understanding, a demand addressed by advances in single-particle cryogenic electron microscopy. This innovative technology has furnished critical insights into ABC transporter architecture, supramolecular assemblies, and mechanistic intricacies [127]. 

Adding another layer to the intricacies of cancer treatment, the secreted phosphoprotein 1 (SPP1), also known as osteopontin (OPN), exhibits upregulation in malignancies, correlating with treatment resistance [128]. Treatment with exogenous OPN amplifies the expression of ABCB1 and ABCG2 transporters, consequently heightening resistance to topotecan and DOX [129].

Furthermore, an overexpression of ABCG2, also known as Breast Cancer Resistance Protein (BCRP), in tumors contributes significantly to drug resistance. ABCG2, an ATP-binding cassette efflux transporter, has been implicated in conferring resistance to clinically used anticancer drugs, including topotecan and irinotecan [130,131]. Inhibitory interventions, such as elacridar and tariquidar, small-molecule inhibitors, exhibit promise by inducing synergistic apoptosis and heightened drug sensitivity in small cell lung cancer (SCLC) cells resistant to VP-16 or SN-38 [132].

BCRP, categorized as an ABC half-transporter, is notably overexpressed in cancer cell lines treated with topotecan or mitoxantrone (MXT), leading to resistance against camptothecin (CPT) derivatives like irinotecan and SN-38 [133,134].

Although the potential of ABC transporter modulators to enhance the efficacy of anticancer drugs has been recognized, the development of MDR1 as a therapeutic target has encountered setbacks [13,135,136]. However, recent work in pharmacophore enhancement, tumor uptake, and carrier–drug association, exemplified by the hydrolytically activatable PF108-[SN22]2, present a promising therapeutic strategy for patients grappling with multidrug-resistant disease [137,138]. A summarized flow model is presented in Figure 1 [135,136,137,138].

## 8. GSH Depletion and Topo-Drug Resistance

Glutathione (GSH) emerges as an important molecule in the context of cytotoxicity induced by topo-active drugs, comprising a tripeptide structure composed of glutamate, cysteine, and glycine. The intricate involvement of GSH extends across various cellular processes, encompassing cell differentiation, ferroptosis, apoptosis, proliferation, and anticancer responsiveness [19].

Recent research illustrates the multifaceted regulatory role of GSH, with perturbations in its concentration linked to tumor genesis, progression, and treatment response. Notably, elevated GSH concentrations within neoplastic cells correlate with heightened chemoresistance. Molecular modifications in the GSH antioxidant system and disruptions in GSH homeostasis establish critical links to resistance against cancer therapeutics [139,140]. 

Among the various molecular mechanisms potentiating resistance to topo-active drugs in cancer cells, GSH depletion and reactive oxygen species (ROS) elevation leading to oxidative damage are observed. Depletion of GSH and elevation of ROS induce oxidative stress, contributing to resistance against topo-active drugs. Inhibitors of GSH biosynthesis reduce the effects of the topo-active drug VP-16, while N-Acetylcysteine, a GSH synthesis promoter, enhances the anticancer effects of topo-active drugs [141,142,143].

Nano-particle-loaded GSH proves effective in reducing resistance to drugs such as DOX and CPT in cancer cells [144,145,146,147]. The topo-active drug CPT, known to induce ROS formation, sees its sensitivity modulated by GSH, which detoxifies these ROS, mitigating drug-induced oxidative stress. Topotecan, inducing both ROS formation and GSH–drug conjugation, exhibits synergistic cytotoxicity with ascorbic acid [148,149,150]. 

Reduced GSH-mediated oxidative stress levels contribute to topotecan-mediated cell death in liver cancer cells, emphasizing the role of GSH in mitigating drug-induced cytotoxic effects. Antioxidants like neobavaisoflavone are proposed to enhance GSH depletion, potentially potentiating the effects of DOX. Irinotecan exacerbates hepatic oxidative stress, lowering GSH levels in cancer cells [32,151]. 

Recently, some data have indicated that irinotecan may worsen hepatic oxidative stress by inducing the formation of ROS and lipid peroxides while simultaneously lowering GSH, superoxide dismutase (SOD), and catalase (CAT) in cancer cells [152]. As topo-active drugs decrease the level of GSH, newer derivatives with redox-sensitive targeted delivery which will maintain the GSH level in treated cancer cells are recommended to avoid drug resistance and desirable apoptotic cell death [153,154]. 

The development of multidrug resistance (MDR) is associated with the maintenance of H_2_O_2_ and GSH levels. Monitoring GSH levels emerges as an efficient strategy for detecting drug resistance and guiding patient responses to therapy. Small molecules like IND-2 (4-chloro-2-methylpyrimido [1″,2″:1,5] pyrazolo [3,4-b] quinolone), by modulating ROS levels and inhibiting TOPII, demonstrate the potential to induce apoptotic cell death in prostate cancer cells [143,155,156].

In the context of oxidative stress, the enzyme phosphodiesterase 10A (PDE10A) is elevated in cancer cells, hydrolyzing cAMP and cGMP. The inhibition of this enzyme is linked to the proliferation of tumor cells. PDE10A deficiency or inhibition reduces the apoptosis, malfunction, and atrophy caused by DOX. Recent findings suggest that inhibiting PDE10A attenuates DOX-induced cardiotoxicity and prevents cancer growth, and thus could be a promising strategy in cancer therapy [157,158].

GSH is important in the protection against tumor microenvironment-related aggression, apoptosis evasion, colonizing ability, and multidrug and radiation resistance. Increased levels of GSH and resistance to chemotherapeutic agents have been observed for various anticancer drugs, including platinum-containing compounds, alkylating agents, anthracyclines, and arsenic [141]. Overall, the role of GSH in topo-active drug cytotoxicity and resistance is intricate and context-dependent, necessitating a detailed understanding for the development of more effective cancer therapies and strategies to overcome drug resistance.

## 9. DNA Damage Response Pathways and Topo-Active Drug Resistance

Various malignancies exhibit chemotherapeutic resistance due to DNA integrity and replication issues. This resistance, attributed to DNA damage, especially that arising from the inhibition of topoisomerase II (TOPII), allows the persistence of damaged cell populations. Homologous recombination (HR) and non-homologous end joining (NHEJ) are primary repair pathways for double-strand breaks (DSBs), with nucleotide excision repair (NER) addressing lesions causing structural distortions in the DNA double helix [159,160,161,162].

However, TOPII inhibitors create DNA adducts, inter-strand crosslinks, and ROS-induced DNA damage, with some evidence reporting the involvement of alternative repair pathways in removing chemotherapeutic drug damage [163,164,165,166,167]. Notably, tyrosyl-DNA phosphodiesterase plays a crucial role in reconnecting and regenerating damaged DNA fragments, forming a covalent intermediate with a tyrosine residue. Moreover, topoisomerase-associated DNA replication forks, particularly TOP3α, are linked to genomic instability, emphasizing the intricate connection between DNA replication, repair, and drug-induced damage [168,169,170].

Topo-active drugs are genotoxic, causing significant DNA damage that triggers a robust repair response pathway. This process potentially plays a role in the development of drug resistance in treated cancer cells. Therefore, combinations of small molecular inhibitors against specific DNA repair proteins, such as PARP, ATM, Rad51, and MGMT, with cytotoxic topo-active drugs prove promising for enhanced anticancer efficacy. Previous studies have indicated the role of MGMT in the emergence of drug resistance to irinotecan, SN-38, and DX-8951f. The elevated expression of the MGMT gene decreased drug sensitivity, while the inhibition of MGMT resulted in an enhanced sensitivity of cancer cells to topo-active drugs [171,172].

Topo-active drugs like topotecan and methotrexate (MTX) cause DNA strand breaks that, in turn, phosphorylate histone H2AX, which may not involve double-stranded DNA breaks. However, double-strand DNA breaks are formed during DNA repair, and analyses of H2AX phosphorylation may be indicative of the extent of the repair process [173]. Data also indicate the involvement of DNA repair proteins, e.g., a mutation in BRCA1 may be related to the drug responsiveness of topo-active drugs. Hence, combinations of small molecular inhibitors against specific DNA repair proteins and cytotoxic drugs such as topo-active drugs could be combined to achieve successful combinatorial anticancer drug approaches [174,175]. Topotecan- and MXT-dependent double-strand breaks also induce activation of DNA repair proteins, ATM and Chk2, in human lung adenocarcinoma A549 cells [176,177]. And combinatorial approaches using small-molecule inhibitors of ATM and Chk2 with topo-active drugs, therefore, were investigated as a novel anticancer approach for better sensitivity and reduced toxicity.

Evidence of the combinatorial approach is further supported by the combined effects of the tyrosine kinase inhibitors erlotinib or gefitinib with CPT that resulted in increased anticancer effects in breast cancer cells [178]. Additional research indicated that the combinatorial role of TOPII inhibitors (idarubicin, daunorubicin, MXT, VP-16) and selinexor (a selective inhibitor of XPO1) was more effective in acute myelogenous leukemia [179].

Notably, andrographolide analog 3A.1, a novel TOPII inhibitor, induces apoptosis by cleaving DNA repair protein PARP-1. Designing dual inhibitors targeting both PARP and topoisomerase, such as 4-amido benzimidazole acridines, enhances cell death in breast cancer cells [180,181,182]. The exploration of concomitant targeting of tyrosyl-DNA phosphodiesterase enzymes and TOPI/TOPII reveals potential avenues for anticancer agents [183].

Emerging combinatorial approaches involving platinum (II) complexes, e.g., [PtCl(NH3)2(9-(pyridin-2-ylmethyl)-9H-carbazole)]NO3 (OPPC), [PtCl(NH3)2(9-(pyridin-3-ylmethyl)-9H-carbazole)]NO3 (MPPC), and [PtCl(NH3)2(9-(pyridin-4-ylmethyl)-9H-carbazole)]NO3 (PPPC), targeting the DNA repair proteins Ku70 and TOPII were effective [184]. Additionally, 2-Amino-Pyrrole-Carboxylate (2-APC) amplifies doxorubicin (DOX) cytotoxicity by inhibiting DNA damage repair via Rad51 recombinase reduction. These intricate combinatorial strategies showcase the evolving landscape of anticancer research, emphasizing the interconnected roles of DNA repair proteins and topo-active drugs [185,186].

## 10. Metabolic Reprogramming and Topo-Active Drug Resistance

Cancer is characterized by a profound shift in metabolism, a phenomenon known as metabolic re-programming. This intricate process encompasses increased lipid metabolism, glutaminolysis, glycolysis, mitochondrial biogenesis, and activation of the pentose phosphate pathway, among other metabolic alterations. In addition, these adaptations furnish cancer cells with essential metabolites, enabling substantial biosynthesis, sustained proliferation, and pivotal processes in tumorigenesis [35]. One exemplification of this metabolic shift is the Warburg effect, a preference for glycolysis and lactate production even in the presence of oxygen, indicating cell-autonomous regulation by oncogenes in numerous proliferating cancer cells and tumors [37].

Petrella et al.’s findings revealed metabolic changes following VP-16 treatment, showcasing an immediate glycolysis arrest succeeded by a gradual recovery marked by epithelial–mesenchymal transition (EMT) and repopulation. Intriguingly, oxidative phosphorylation (OXPHOS) initially surged but subsequently decreased, persisting above control levels. These insights shed light on the energy metabolism dynamics during topo-active drug therapy, unveiling potential pharmacological targets for castration-resistant prostate cancer (CRPC) treatment, which presents a distinct scenario [187].

Metabolic processes, responsible for converting nutrients into energy and crucial cellular chemicals, play a pivotal role in cancer development. The PI3K/AKT and MAPK signaling pathways exert significant influence on both tumor progression and immunity. Simultaneous suppression of metabolic pathways, such as glycolysis and PI3K/AKT/mTOR, halts tumor development. In particular, immune cells undergo shifts in energy consumption, while tumor cells adapt their metabolism to fuel their growth. This creates a competitive environment for nutrients between tumors and immune cells, thereby influencing immune responses. The development of dual signaling pathway inhibitors targeting PI3K/AKT/mTOR and other pathways is underway to enhance cancer therapy [188,189].

Frequently, cancer cells undergo altered metabolic states, increasing tumor aggressiveness [190]. Clinical studies targeting these metabolic pathways show promise, but success hinges on meticulous consideration of toxicity and strategic implementation of combination therapy. In the pursuit of common metabolic reprogramming pathways in cancer cells, a note of caution is essential when employing metabolic regulators, given their potential for toxicity concerns [191].

## 11. Immune Checkpoint Inhibitors and Topo-Active Drugs

Checkpoint inhibitors (CPIs) have emerged as a pivotal component in immunotherapy, revolutionizing the treatment of various cancers. The intricate interplay between the immune system’s checkpoints and cancer cells is a critical focus, as malignant cells exploit these checkpoints to evade detection. CPIs effectively disrupt these interactions, allowing the immune system to discern and eliminate cancer cells with heightened efficacy. This therapeutic approach is particularly promising for metastatic and chemotherapy-resistant malignancies [192].

In the tumor microenvironment (TME) of numerous human cancers, there is a prevalent upregulation of immune checkpoints and their responsive ligands, acting as formidable barriers to initiating an effective antitumor immune response [193,194]. The primary strategies for checkpoint blockades involve targeting the interaction between programmed cell death 1 (PD-1 or CD279) and programmed cell death ligand 1 (PD-L1 or CD274 or B7 homolog 1) and inhibiting cytotoxic T-lymphocyte-associated protein 4 (CTLA-4 or CD152) [195].

During the immune-response-induction phase, co-stimulation of CD80/CD86 via CD28 provides crucial stimulus signals for T-cell proliferation and efficient differentiation [196]. The ligands for PD1 (PDL1 and PDL2) are expressed in both cancer cells and antigen-presenting cells [197]. PD1, an immune-cell-specific surface receptor, lowers the apoptosis threshold, dampens T-cell receptor signaling, induces T-cell exhaustion, and depletes T cells in the presence of a ligand [198,199]. Upregulated PDL1 expression in some tumor cells enhances T-cell suppression, favoring tumor cell survival. Monoclonal antibodies against PD1 disrupt the PD1-PDL1 link, increasing the immune system’s response and halting tumor progression [200,201].

In the realm of immunotherapy, the potentiation of antitumor therapy is exemplified by blocking PD-1 or PD-L1 using various inhibitors, including anti-PD1 antibodies [202]. Combining topo-active drugs such as SN-38 with anti-PD1 antibodies is being explored at both preclinical and clinical levels [203,204]. Drug conjugates, like combining a TDO inhibitor with irinotecan, showcase potential in inhibiting TDO enzyme activity and inducing apoptosis in cancer cells [205,206].

Clinical trials utilizing antibody–drug conjugates targeting Trop-2, like Sacituzumab govitecan and SN-38, have been reported, with a focus on breast cancers [207,208]. The antibody–drug conjugates are explored as drug payloads against TOPI and target selective tumor antigens, predominantly TROP-2. Emerging evidence supports the link between chemotherapeutic drugs, as topo-acting drugs such as MTX may upregulate major histocompatibility class I (MHC-I) antigen processing and presentation [209,210]. In turn, propositions of the combinatorial approaches that can combine topo-acting drugs and immune checkpoint inhibitor therapies (ICIT) such as the PD-1/PD-L1 [PD-(L)1] blockade are warranted [209,210,211,212]. Another topo-active drug, SN-38, has shown increased efficacy in immune checkpoint blockade therapies in a syngeneic tumor model [204,213].

An interesting observation suggested that the use of the chemo-immunotherapeutic approach, combining the topo-active drug teniposide with anti-PD1 antibodies, shows promise for enhanced antitumor efficacy in mouse tumor models, with implications for the STING pathway [214,215]. Recent emphasis on antibody–drug conjugates, like trastuzumab-L6 and the DNA TOPI inhibitor MF-6, highlights their role in achieving immunogenic tumor cell death [216,217].

Pancreatic cancer is highly resistant to topo-active drugs due to their ability to induce intrinsic physical and biochemical stresses, causing an increase in interstitial fluid pressure, vascular constriction, and hypoxia. Immunotherapies, e.g., therapeutic vaccines, immune checkpoint inhibition, CAR-T cell therapy, and adoptive T-cell therapies are also ineffective against pancreatic cancers due to their highly immunosuppressive nature. This drug-resistant and immunosuppressive nature could be overcome by the development of nanocarrier-based drug formulations [218]. PEGylation, the conjugation of polyethylene glycol (PEG) to drugs, has been employed in Doxil and Genex-ol-PM drug formulations that are FDA-approved nanocarrier drug formulations against PDAC and metastatic PDAC [219]. The use of PLGA-NPs coated with taxol led to 90% drug release, leading to decreased pancreatic tumor volume as compared to controls [220]. Cationic poly-lysine NPs coated with HIF1alpha siRNA and GEM (gemstones) are used in siRNA-adjuvant GEM therapy in the treatment of pancreatic cancers. The chemo-resistant and invasive nature of pancreatic CSCs is mediated by miRN,A resulting in the manifestation of cancer recurrence and drug resistance [221]. Micelles conjugated with GEM and miRNA mimetics were tested against pancreatic CSCs. The synergistic effect of GEM and miRNA mimetics showed decreased tumor growth in CSCs, GEM-resistant CSCs, and xenograft pancreatic cancer [222].

In conclusion, checkpoint inhibitors represent a remarkable advancement in cancer treatment, notably for melanoma, lung cancer, and kidney cancers (Figure 2). Despite their notable efficacy, CPIs are not without drawbacks, as immune-related toxicities can arise from an overactivation of the immune system, necessitating vigilant monitoring for potential side effects.

## 12. Topo-Active Drugs and CSCs

CSCs, also referred to as tumor-initiating cells, and crucial for tumor development, growth, and resistance, can emerge under therapeutic pressure and altered microenvironments. CSCs can be generated when therapeutic pressure and an altered microenvironment are present and formed from non-CSCs or senescent tumor cells following therapy [223]. Compared to non-CSCs, CSCs are distinct for their immune resistance, and cancer immunosurveillance enhances certain populations of cancer cells with stem-like characteristics. For CSCs to maintain the tumorigenic process, CSC immune evasion is essential, including elevated PD-L1 expression [224,225]. In addition to immune evasion, CSCs display plasticity in terms of favorable DNA repair pathways in response to genotoxic-mediated DNA damage. Therefore, understanding CSC pathways influencing resistance against topo-active drugs is crucial for designing effective combinational anticancer therapies.

Recent studies identify drugs, such as aloe-emodin and digoxin, displaying anticancer and anti-CSC properties. In vitro and in vivo analysis, notably using lung cancer PDX models, revealed that the single-cell transcriptomics analysis of digoxin could suppress the CSC subpopulation and cytokine production in CAF-cultured cancer cells [226]. Moreover, combinatorial approaches involving digoxin and topo-active drugs may enhance therapeutic outcomes, addressing drug resistance linked to topo-active agents.

Standard cancer therapies often fail due to the generation of CSCs, with signaling pathways like Wnt, Hedgehog, Notch, hypoxia and PI3K/AKT/mTOR being aberrantly regulated in these cells [225,226,227]. These signaling pathways are also involved directly or indirectly with resistance to topo-active drugs in cancer cells [33]. In this context, mesenchymal stem cells (MSCs) are reported to show adjustability to TOPI and TOPII inhibitors even after being exposed to high dosages of irinotecan or VP-16, suggesting their potential as therapeutic targets for chemotherapy-induced bone marrow damage [228]. TOPII gene expression varies between CSCs and non-CSCs in the glioma with higher levels present in CSCs. Silencing of TOPII has been shown to induce cell cycle arrest and apoptosis in CSCs [229]. This link between resistance and topoisomerase is supported by the fact that dual treatment with CPT-containing nanoparticle–drug conjugates and CRLX10, an inhibitor of HIF-1α, resulted in enhanced efficacy in breast cancer cells [229,230,231].

Topo-active drug resistance is a significant problem in the treatment of cancer because the cancer cells develop mechanisms against therapeutic drugs, giving rise to more aggressive and fit clones that worsen treatment. There are several CSC clones already present, and some of them can quickly adapt to changes in the TME and/or grow when exposed to radiation and chemotherapy [223]. Targeting CSC subpopulations has been suggested to eliminate tumors and prevent their recurrence. Both ROS and RNS (reactive nitrogen species) are expertly managed by CSCs, and they use the TME to their benefit. CSCs also have improved DNA repair capability, and the ability to turn off apoptotic pathways, leading to drug resistance [223,228,232]. As a result, combating CSC drug resistance by using the proper medications to block both TOPI and TOPII seems logical [233].

Intriguingly, recent developments in the field suggest that the utilization of topo-active drugs, such as DOX and VP-16, in conjunction with an autophagy inhibitor, may lead to a favorable induction of cell death in drug-resistant CSCs (Figure 3) [33] [224,228,234]. In addition, recent studies indicate that the deregulation of TOPIIα is strongly linked to the growth and progression of CSC features with the involvement of Laminin-332 and YM-1 [224]. DOX-resistant non-small cell lung cancer (NSCLC) exhibited stem-related markers CD133 and OCT4 and showed strong canonical Wnt activity over DOX-sensitive cells [235].

## 13. Epigenetic Changes and Topo-Active Drugs

Alterations in chromatin structure are evident in cancer-associated histone mutations, leading to chromatid remodeling, heightened histone exchange, and nucleosome displacement. These changes, alongside epigenetic modifications, including DNA methylation, histone acetylation, and RNA interference, play pivotal roles in cancer development, progression, and drug resistance [236,237,238,239,240].

In the epigenomic remodeling process, histone modification involves changes to histones, the proteins around which DNA strands coil, encompassing the addition or removal of acetyl groups. The addition of acetyl groups, referred to as histone acetylation, serves to activate the corresponding chromosomal region, while their removal (deacetylation) leads to deactivation. Key enzymes in the regulation of gene expression are histone deacetylases (HDACs) [12,239].

The dynamic equilibrium of acetylation levels on histone-conserved lysine residues, which control chromatin remodeling and gene expression, is maintained by them. The connection between cancer and abnormal HDAC activity has been widely documented, and the inhibition of human tumor cell line proliferation by HDAC inhibitors has been demonstrated in vitro. Moreover, potent antitumor activity in human xenograft models has been observed with several HDAC inhibitors (HDACis), suggesting their potential as new cancer therapeutic agents [241,242].

In a typical cellular environment, HDACs play a role in promoting cell cycle progression by repressing gene transcription at the promoter level and direct deacetylation of important regulators of the cell cycle, and also contribute to the development of cancer drug resistance. Novel strategies could be employed to enhance the therapeutic effectiveness of HDACis, potentially resulting in improved, targeted classes of drugs with reduced adverse effects. FDA-approved inhibitors like vorinostat and depsipeptide have demonstrated efficacy and safety in treating hematologic malignancies. However, treating both hematologic and solid tumors will likely involve combining HDACis with genotoxic drugs such as topo-active drugs [241,243,244].

VP-16-resistant lung cancer cells have been shown to display elevated levels of histone deacetylase 4 (HDAC4) and concomitant treatment with both trichostatin A (an HDAC inhibitor) and VP-16 resulted in enhanced cell death in lung cancer cells [245]. Furthermore, combinations of the pan-HDAC inhibitor panobinostat and topo-active drugs have demonstrated increased effects in cervical cancer cells [246]. Results suggested that the combination of the HDACi vorinostat with the DNA-damaging agent Topo I-active drug SN-38 may also be beneficial and could result in enhanced cytotoxic effects [247,248].

The idea to develop small-molecule agents that can work as dual inhibitors of both HDAC and TOPI/TOPII has been explored [249,250,251,252,253,254,255,256,257]. Kim et al. have also emphasized the importance of dual targeting by the combinatorial treatment of CI-994, an HDAC inhibitor, with VP-16, leading to synergistic anticancer effects through Topo II and Ac-H3 regulation [258]. Studies on agents containing pyrimido [5,4-b] indole and pyrazolo [3,4-d] pyrimidine motifs as novel compounds indicated the relevance of dual inhibitors of HDAC and Topo II and such combinations have been suggested as a new avenue for enhanced anticancer responses [259].

Thus, epigenetic modifications, specifically DNA methylation and histone acetylation, influence cancer cells to acquire resistance against various topo-active drugs [248]. Modification leading to hypomethylation can either increase or decrease the sensitivity of cells to topoisomerase inhibitors due to altered mechanisms. Also, it has been reported that the dysregulation of histone acetyltransferase (HAT) and histone deacetylases (HDACC), which play a key role in regulating the level of acetylation in histone proteins, results in resistance against topoisomerase inhibitors (Figure 4) [12,248,249,250,251,252,253,254,255,256,257,258,259,260].

## 14. Miscellaneous Mechanisms of Resistance and Topo-Active Drug-Induced Senescence and Drug Resistance

Genotoxic drug-induced senescence is one of the many observed changes in cancer cells that correlate with drug responsiveness to various precision and targeted inhibitors, including topo-active drugs, immune checkpoint inhibitors, DNA repair protein inhibitors, and kinase inhibitors [260]. Drug-induced senescence is often related to distinct forms of cell cycle arrest, telomere instability, and overall genomic instability.

As G2-M-phase cell cycle checkpoints play significant roles in topo-active-drug-mediated responsiveness, the inhibitor UCN-01 has been reported to modulate the activity of p53, which in turn potentiates toxicity in cancer cells [261]. Furthermore, a different type of TOPI inhibitor, imidazoacridinone C-1311, and the TOPII inhibitor SN 28049 have been reported to induce growth arrest and senescence in human lung cancer cells [262,263]. Further studies have also shown that the TOPI inhibitor SN-38 can cause G2 arrest and lead to senescence via downregulation of Aurora-A kinase [264,265].

The relevance of the chromosome instability in cancer cells caused by topo-active drugs such as the TOPII inhibitor ICRF-193 was reported to be due to telomere dysfunction and cellular senescence [66]. Taschner-Mandl et al. have emphasized the action of low-dose topotecan to induce proliferative arrest and a typical senescence-associated secretome in cancer cells [266]. Hao et al. have also shown that TOPI inhibitors such as irinotecan may lead to an elevation of drug-mediated senescence via the cGAS pathway [267]. Topo-active drugs such as irinotecan modulate p53 activity and induce the acetylation of several lysine residues within p53, leading to drug-induced senescence [268]. The sequential combination of therapy-induced senescence and ABT-263 could shift the response to apoptosis. The administration of ABT-263 after either VP-16 or DOX also resulted in marked, prolonged tumor suppression in tumor-bearing animals. These findings support the premise that senolytic therapy following conventional cancer therapy may improve therapeutic outcomes and delay disease recurrence [269].

Mercaptopyridine oxide, a new class of TOPII inhibitor, also demonstrated the activation of G2/M arrest and cellular senescence [270,271,272]. Recent work has shown that DOX treatment of cancer cells leads to the enrichment of miR-433 into exosomes, which in turn induces bystander senescence [273]. Topo I inhibitors such as SN-38 and irinotecan have shown their abilities to induce G2 arrest and cellular senescence via modulation of the activities of p21, p53, Bcl-xL, Bcl-2, and PGC-1-related coactivator [254,273,274]. These modulatory effects are associated with the drug responsiveness and potential relapse of tumors in cancer patients treated with topo-active drugs [275,276,277,278,279,280,281,282].

### 14.1. Regulatory RNAs and Topo-Active Drugs

The interfering role of microRNAs and long noncoding RNAs (lncRNAs) is suggested to modulate the levels of TOPI and TOPII enzymes that, in turn, are related to topo-active drug resistance in cancer cells [12,283,284,285,286]. The regulatory role of microRNAs and lncRNAs is extensively reported and is associated with oncogenic as well as tumor suppressor roles in cancer cells. Previous studies have shown that the downregulation of TOPI observed in hepatic cancer cells is due to an overexpression of miR-23a and this may be one of the factors for drug-induced effects in cancer cells [3]. miR-627 has been suggested to work by inhibiting the intracellular drug metabolism of irinotecan, inducing a better responsiveness [287]. Recent findings have suggested that miR-9-3p and miR-9-5p could contribute to VP-16 resistance in cancer cells by influencing the reduction in TOPII protein levels [288].

Another study investigated the role of miR-21-5p in drug resistance in colorectal cancer cells. High levels of miR-21-5p were found to be associated with poor prognosis in colorectal cancer patients. Overexpressing miR-21-5p in DLD-1 colorectal cancer cells led to drug resistance against topoisomerase inhibitors. The resistance was attributed to reduced apoptosis and increased autophagy without affecting topoisomerase expression or activity. Bioinformatics analysis revealed genetic reprogramming and downregulation of proteasome pathway genes in response to miR-21-5p overexpression. These findings have provided valuable insights into the development of drug resistance in colorectal cancer and suggest potential clinical strategies to overcome it [287,288].

### 14.2. EMT and Topo-Active Drugs

The elevation of epithelial-to-mesenchymal transition (EMT) during topo-active drug resistance has been observed as one of the many changes within the TME. Limited studies are available to decipher the molecular basis of EMT and topo-active drug resistance. A finding suggested that the EMT genes ZEB1 and CDH2 are upregulated in colorectal cancer cells following the development of resistance to DOX [289]. The role of EMT as an enhancer of DOX-mediated drug resistance is indicated via mediators such as ZEB proteins, microRNAs, Twist1, and TGF-β. Combinational uses of small-molecule inhibitors of EMT and topo-active drugs have been proposed as one of the newer approaches for anticancer therapy [290]. TOPII is demonstrated to induce EMT by using a TCF (T-Cell Factor) transcription inducer. To achieve combinatorial avenues, inhibitors of TCF with topo-active drugs have been explored as a new class of therapy [291,292]. Maximum tolerable dosing (MTD) of topotecan is suggested to cause enhanced EMT while extended exposure to topotecan induces increased long-term drug sensitivity by decreasing malignant heterogeneity and preventing EMT [293].

The role of epidermal growth factor receptors in topo-active drugs in cancer drug resistance has also been examined. A study indicated that epidermal growth factor receptor tyrosine kinase inhibitors (EGFR TKIs) like gefitinib and erlotinib combined with TOPI inhibitors could be a promising treatment option for resistant NSCLCs [294]. Osteoprotegerin (OPG) has been reported to be associated with various cancer types, including bladder carcinoma, gastric carcinoma, prostate cancer, multiple myeloma, and breast cancer [295]. High levels of OPG secretion were associated with metastasis and drug resistance in aggressive breast cancer treated with topo-active drugs.

The complicated metabolism of irinotecan makes it a potential target for drug interactions, and it was shown that the UGT 1A1-mediated glucuronidation of SN-38 to SN-38G resulted from the elevated levels of UGT at both the protein and mRNA levels. Overexpression of UGT at both the mRNA and protein levels can be responsible for SN-38 resistance in a human lung cancer cell line [51]. Studies have indicated that irinotecan therapy activates nuclear factor kappa B (NF-kB), a ubiquitous transcription factor, with the elevation of TNF-alpha and oncogenic Ras and the emergence of drug resistance in cancer cells [296].

The modulation of TOPI expression via the ubiquitin/26S proteasome pathway is suggested for TOPI drug resistance. Data have indicated that the formation of ·NO/·NO-derived species in cancer cells can induce a reduction in the level of TOPI via the ubiquitin/26S proteasome pathway [64]. Exploring combinatorial approaches by combining proteasome inhibitors such as MG132 and lactacystin along with topo-active drugs including SN-38 have been reported to potentiate anticancer effects in cancer cells [261,297,298].

## 15. Conclusion and Future Developments

As we have gained a better understanding of the mechanisms involved in the formation and progression of cancers, significant progress has been achieved in the treatment of human cancers in the clinic. Unfortunately, chemotherapy resistance has also remained a challenge both in the treatment of patients and in the discovery of novel cancer drugs. One main aim of this review was to describe the various mechanisms of resistance to topoisomerase-based therapies, and we also believe that with a better understanding of these mechanisms, topoisomerase drugs can be discovered for the treatment of patients in the clinic in the future. The development and synthesis of novel topoisomerase-active compounds may lead to the discovery of new topoisomerase inhibitors with improved efficacy and safety profiles. Recent advances in understanding tumor transformation and progression mechanisms and their regulation by proteins have led to a new era in drug formulation and clinical evaluation. Research on developing drugs that target specific types of topoisomerases could show promising results by reducing off-target effects with better treatment accuracy and reduced side effects. Also, the use of natural compounds from various organisms or plants needs to be explored for their unique structure and mechanism of action as novel topoisomerase inhibitors.

Recent advances in drug delivery system(s) also offer exciting opportunities for packaging active topo-active drugs. Significant progress has been made in the field of encapsulation of chemotherapeutics and the FDA has now approved various liposomes for therapy which should be tried with topo-active drugs. Another area that needs to be explored is utilizing antibody–drug conjugate delivery systems as they are highly specific for the precise delivery of drugs to tumors. While exosomes for the delivery of chemotherapeutics are not well studied, recent advancements would suggest chemotherapeutic drugs like DOX and SN-38 can also be packaged into exosomes for the treatment of human cancers. The discovery of nanoparticles plays a key role in improved drug formulation by enhancing the delivery, solubility, and stability of the drug. More research on nanoparticle-based cancer drugs helps in developing sustained release and targeted delivery of topo-drugs, which can eventually reduce the side effects of these drugs and can also prevent the development of drug resistance.

The implementation of combinatorial therapies by combining a topoisomerase inhibitor with another type of drug in a single formulation can improve efficacy and simplify treatment regimens. Researchers and clinicians must develop combination therapies or newer drugs that target multiple mechanisms simultaneously or employ strategies to reverse or prevent resistance. Although combinatorial therapy is complex, it is recommended to conduct more investigations on different combinations of drugs for better treatment and to overcome drug resistance in the clinic. Our recent research has shown that nitric oxide-generating drugs, e.g., J-SK or NCX4040, are highly effective in inhibiting the functions of ABC transporters, resulting in the sensitization DOX and topotecan in ABC-transporter-expressing cells [299,300]. We believe that this needs to be further explored in a clinical setting as these combinations may be highly beneficial for the treatment of patients overexpressing an MDR phenotype. Another active area of research is the use of combinations of ferroptosis inducers, e.g., Erastin or RSL3, with anticancer drugs to overcome resistance. We have found that such combinations of ferroptosis inducers such as erastin with the topo-active drug DOX result in an overcoming of drug resistance in Pgp-overexpressing tumor cells (Sinha et al., unpublished observations).

Treatments have become precise, personalized, and targeted toward each patient, and have been widely adopted in clinical practice due to the development of techniques like next-generation sequencing (NGS), comparative proteomics, structural biology, and computational methods. As genetic mutations, chromosomal abnormalities, epigenetic alterations, and extracellular vesicles (EVs) play significant roles in the growth of tumors, these also represent a vast untapped resource of knowledge with enormous potential as cancer biomarkers in precision medicine. By sequencing a patient’s genetic information, it is possible to identify any abnormal gene expressions and genetic aberrations, leading to rapid diagnosis and targeted therapy.

As precision medicine has become essential, it also has its disadvantages such as the cost of the test, reachability among the population, and ethical concerns. Therefore, for successful precision therapy, it is necessary to obtain additional knowledge on comparative genomics, transcriptomics, and proteomics for better understanding. Furthermore, more research should be conducted on a large population model to stop adverse events, eliminate unnecessary and inefficient treatments, and deliver more efficient targeted medicines.

In cancer, understanding the resistance mechanisms of existing drugs is crucial. Currently, the problem of multidrug resistance in cancer is treated by implementing various strategies like combinational therapy, synthetic analogs, precision medicine, and immunotherapy. Eventually, these cancer cells become resistant to new drugs by undergoing genetic alteration, epigenetic changes, and other strategies. Therefore, it is essential to study the resistance mechanisms of cancer for the development of new treatments to overcome or bypass these multidrug-resistant cancer cells.

The development of clinical care for many cancers with poor prognoses has been made possible by the promising emergence and success of cancer immunotherapy. Despite significant advancements in immunotherapies, there still are various difficulties, e.g., low response rates, an inability to anticipate clinical effectiveness, and possible adverse effects such as cytokine storms or autoimmune responses. Therefore, it is recommended to find newer immunotherapeutic agents and immunotherapy technologies for improved treatment with a quick onset of action, reduced toxicity, and increased efficiency.

Although CSC targeting is an interesting area of research in cancer therapy, it is essential to address the root cause of cancer for the treatment of CSCs as they survive much longer than normal cells and accumulate genetic mutations. It is recommended that the implementation of different strategies like epigenetic therapies, personalized medicine, and combinational therapies is further researched for the improvement of treatments. Novel CSC-targeting inhibitors can become potential drugs that disrupt self-renewal and the differentiation of CSCs with less toxicity and more accuracy.

In conclusion, an increased understanding of the mechanisms underlying topo-active drug resistance suggests an integrated approach that mitigates pro-tumor molecular programming such as oxidative stress, metabolic reprogramming, DNA repair response, epigenetic remodeling, EMT, and CSCs (Figure 5). The recent utilization of molecular targeting agents for these signaling pathways remains an important approach towards the desired success of topo-active drugs. However, the use of these targeted therapies in combination with topo-active drugs is not without limitations, specifically in terms of success at the preclinical and clinical levels. Further research is needed to identify combinatorial approaches to repurpose existing drugs and to explore new classes of drugs that can demonstrate multiple targeting properties in different human cancer types and at the same time minimal side effects in cancer patients.

## Figures and Tables

**Figure 1 cancers-16-00680-f001:**
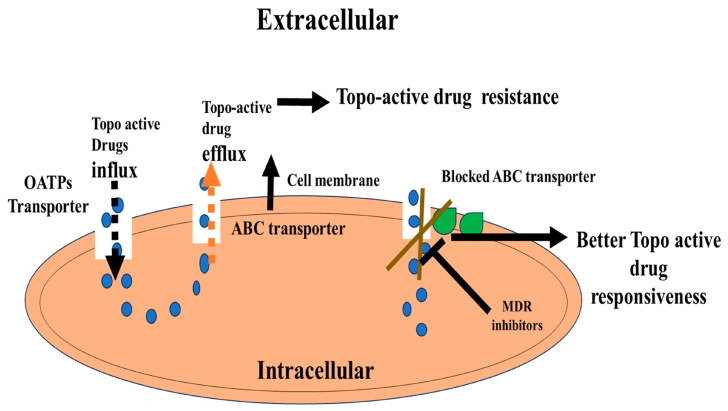
The role of ABC transporter in topo-active cancer drug resistance. Organic anion transporting polypeptide (OATP) transporters for the influx of topo-active drugs. ATP-binding cassette (ABC) efflux transporters such as P-gp are overexpressed in cancer cells and contribute to multidrug resistance (MDR) against topo-active drugs. P-glycoprotein (P-gp).

**Figure 2 cancers-16-00680-f002:**
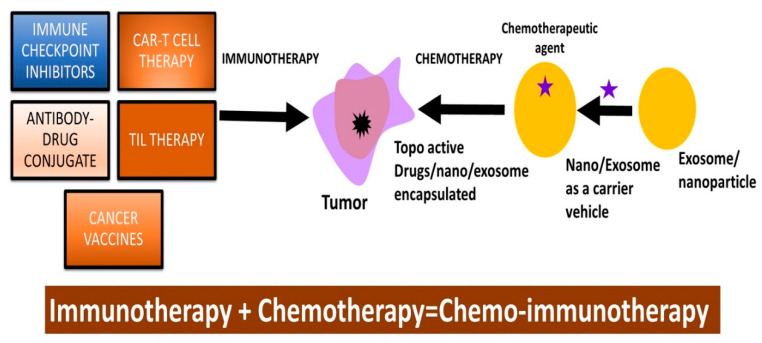
Illustration on the relevance of immunotherapy in combination with topo-active drugs to suppress drug resistance in cancer cells. Tumor-infiltrating lymphocyte (TIL) therapy; topoisomerase-active drugs (topo-active drugs).

**Figure 3 cancers-16-00680-f003:**
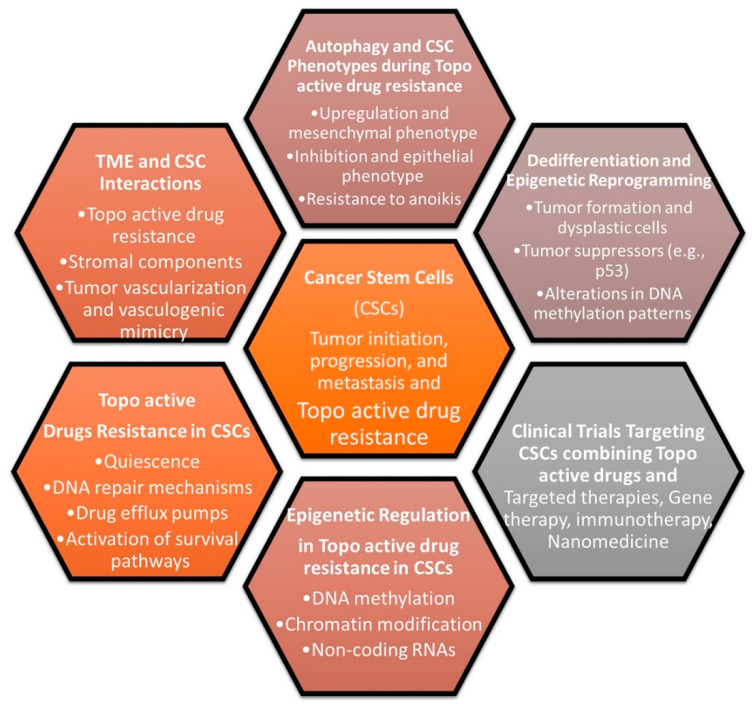
Molecular pathways that involve the role of CSCs in topo-active drug resistance. Cancer stem cells (CSCs), topoisomerase-active drugs (topo-active drugs), and tumor microenvironment (TME).

**Figure 4 cancers-16-00680-f004:**
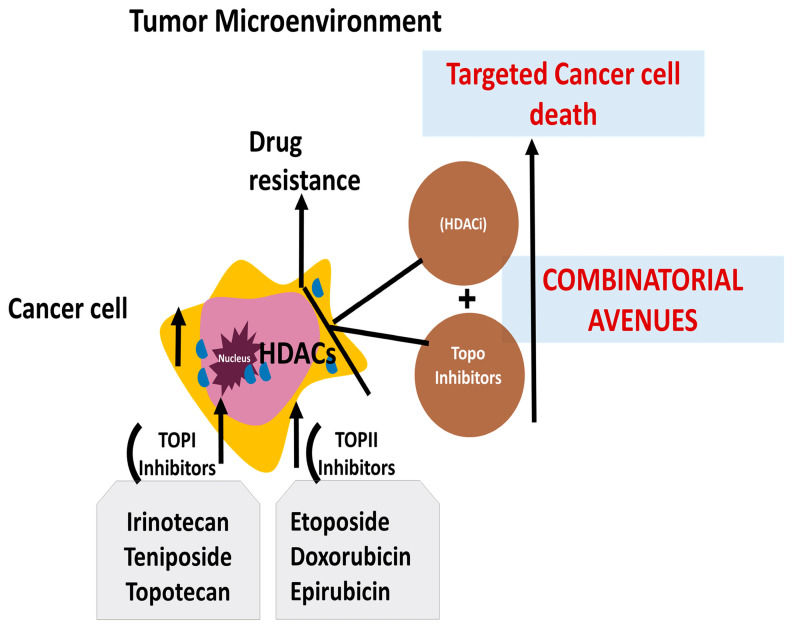
Combinatorial approaches to using epigenetic enzyme inhibitors and topo-active drugs for enhanced drug responsiveness. Topoisomerase I (TOPI), topoisomerase II (TOPII), and topoisomerase inhibitors (topo inhibitors); histone deacetylase inhibitor (HDACi).

**Figure 5 cancers-16-00680-f005:**
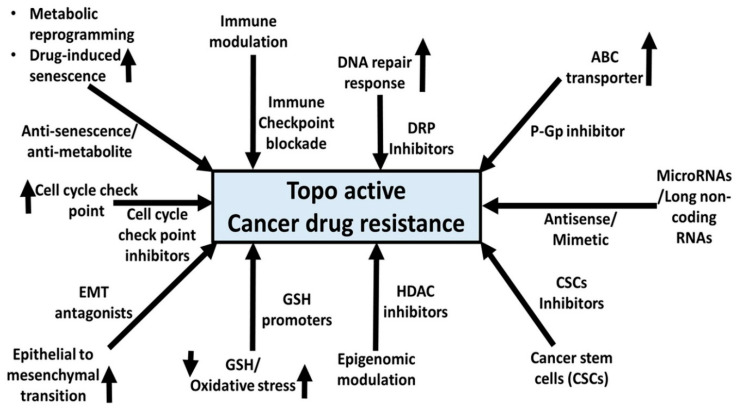
A summarized illustration of various molecular mechanisms and the scope of potential combinatorial avenues that can alleviate topo-active drug resistance in cancer cells. DNA repair protein (DRP) inhibitors. ATP-binding cassette (ABC) transporter. Epithelial–mesenchymal transition (EMT). Glutathione (GSH). Histone deacetylase (HDAC). Cancer stem cells (CSCs).

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
