# Peer review of "Understanding Cancer’s Defense against Topoisomerase-Active Drugs: A Comprehensive Review"

_cancers, 2024, doi:10.3390/cancers16040680_

Round 1

Reviewer 1 Report (Previous Reviewer 1)

Comments and Suggestions for Authors

The authors underwent a review process, during which the quality of the paper significantly improved. However, there is a need to correct the English notation in some sentences, particularly in terms of terminology. 

Overall, accept in present form.

Comments on the Quality of English Language

The authors underwent a review process, during which the quality of the paper significantly improved. However, there is a need to correct the English notation in some sentences, particularly in terms of terminology. 

Overall, accept in present form.

Reviewer 2 Report (Previous Reviewer 2)

Comments and Suggestions for Authors

The paper is suitable for publication now.

This manuscript is a resubmission of an earlier submission. The following is a list of the peer review reports and author responses from that submission.

Round 1

Reviewer 1 Report

Comments and Suggestions for Authors

cancers-2800177

 Type of manuscript: Review

 Title: Cracking the Code: Understanding Cancer's Defense against Topoisomerase-Active Drugs: A Comprehensive Review

 Authors: Nilesh Kumar Sharma, Anjali Bahot, Gopinath Sekar, Mahima Bansode, Kratika Khunteta, Priyanka Vijay Sonar, Ameya Hebale, Vaishnavi Salokhe and Birandra K. Sinha*

 This paper is a highly detailed review of topoisomerase inhibitors as anticancer agents. While overall well-written, there are a few additional points that need to be addressed as highlighted in the following comments.

 [Major concerns]

1. Abbreviations: The use of abbreviations when writing a paper has many advantages besides simplicity of expression. To use an abbreviation, first write the abbreviation in parentheses after the full name, and then use the abbreviation from Introduction to the final Conclusion. Only in Abstract and Figure legend do it separately. In addition, when using abbreviations, use the same abbreviation (Line 41 topo-active vs line 245 TOP, line 118 Topoisomerase II (Top II) vs Line 153, 162 and 163 Top2). If an abbreviation is not used more than twice, there is no need to define it, so please delete it. In the case of an abstract, the use of abbreviations is completely separate from the main text. As you know, there are cases where only the abstract is introduced separately, so in the abstract, abbreviations should only be used if they are repeatedly used and if they are not used again, only the full name should be used.

 * There is no definition of abbreviation. Please define the followings.

1) Line 17: ABC transporters

2) Line 19, 22, 25, 41, 46, 54, 63: Topo-active drugs

3) Line 41, 46, 54, 63: Topo-active drugs

4) Lines 51 and 52: Topo I and Topo II’

5) Line 67: ROS

6) Line 207: MDR1

7) Line 290: DOX

8) Line 293: NADPH

9) Line 324: BBB

10) Line 381: CPT

11) Line 394: SOD and CAT

12) Line 444: PARP and ATM

13) Line 458: Chk2

14) Line 559: PD-1/PD-L1 [PD-(L)1]

15) Line 741: cGAS

16) Line 754: Bcl-xL

17) Line 781: ZEB1 and CDH2

18) Line 784: TGF-β

2. In cases where abbreviations are used within figures, please list these abbreviations along with their corresponding full names in the figure legends. If there are two or more abbreviations, arrange them in alphabetical order.

1) Figure 1: OATP and MDRI

2) Figure 2: TIL

3) Figure 3: Topo and TME

4) Figure 4 : HDAC

5) Figure 5: DRP, ABC transporter, EMT, GSH, and HDAC

3. Line 669: The authors, in the midst of discussing carcinogenesis, abruptly introduced content related to apigenin. Does this align with the overall context of the paper? If necessary, it would be appropriate to place reference 240 after the sentence describing apigenin.

4. Figures 1 through 5 only have titles without accompanying explanations of the content. It would be beneficial to enhance reader understanding by adding descriptions to the figure contents and providing full names for all abbreviations used in the respective figure legends. When listing abbreviations, arrange them alphabetically for clarity.

[Minor concerns]

1. Lines 42 and 43: The font size does not match the format of the manuscript.

2. Reduce the spaces immediately preceding the words mentioned below by removing unnecessary single-letter spacing.

1) Line 146: The unique

2) Line 220: Often

3) Line 443: studies

4) Line 451: double

5) Line 454: combinations

6) Line 536: In

7) Line 575: Drug

8) Line 733: to

9) Line 839: The

3. Lines 318, 370, 650, 760, and 776: References must be listed in single parentheses.

4. Line 510: There should be a period at the end of the reference.

5. Line 190: Taxus brevifolia, the Pacific yew or western yew, is a species of tree in the yew family Taxaceae native to the Pacific Northwest of North America. Therefore, ‘Pacific yew Taxus brevifolia’ should be corrected to ‘Taxus brevifolia, Pacific yew,’ and.......

6. Lines 245, 247, 255, 256, and 257: ‘TOP1’ ‘hTOP2A’ ‘hTOP2B’ ‘hTOP2’ should be used as a uniform abbreviation.

7. Lines 268 and 296: The C in ‘Comptothecin’ should be lowercase.

8. Lines 388 and 399: H2O2 should be written as H2O2.

9. Reference section: Author should consult and peruse carefully recent issues of the journal, Cancers, for format and style. Author must modify all the reference section.

Journal Articles:1. Author 1, A.B.; Author 2, C.D. Title of the article. Abbreviated Journal Name Year, Volume, page range.

Overall, the manuscript can be considered to publication after major revision as indicated above.

Comments on the Quality of English Language

cancers-2800177

 Type of manuscript: Review

 Title: Cracking the Code: Understanding Cancer's Defense against Topoisomerase-Active Drugs: A Comprehensive Review

 Authors: Nilesh Kumar Sharma, Anjali Bahot, Gopinath Sekar, Mahima Bansode, Kratika Khunteta, Priyanka Vijay Sonar, Ameya Hebale, Vaishnavi Salokhe and Birandra K. Sinha*

 This paper is a highly detailed review of topoisomerase inhibitors as anticancer agents. While overall well-written, there are a few additional points that need to be addressed as highlighted in the following comments.

 [Major concerns]

1. Abbreviations: The use of abbreviations when writing a paper has many advantages besides simplicity of expression. To use an abbreviation, first write the abbreviation in parentheses after the full name, and then use the abbreviation from Introduction to the final Conclusion. Only in Abstract and Figure legend do it separately. In addition, when using abbreviations, use the same abbreviation (Line 41 topo-active vs line 245 TOP, line 118 Topoisomerase II (Top II) vs Line 153, 162 and 163 Top2). If an abbreviation is not used more than twice, there is no need to define it, so please delete it. In the case of an abstract, the use of abbreviations is completely separate from the main text. As you know, there are cases where only the abstract is introduced separately, so in the abstract, abbreviations should only be used if they are repeatedly used and if they are not used again, only the full name should be used.

 * There is no definition of abbreviation. Please define the followings.

1) Line 17: ABC transporters

2) Line 19, 22, 25, 41, 46, 54, 63: Topo-active drugs

3) Line 41, 46, 54, 63: Topo-active drugs

4) Lines 51 and 52: Topo I and Topo II’

5) Line 67: ROS

6) Line 207: MDR1

7) Line 290: DOX

8) Line 293: NADPH

9) Line 324: BBB

10) Line 381: CPT

11) Line 394: SOD and CAT

12) Line 444: PARP and ATM

13) Line 458: Chk2

14) Line 559: PD-1/PD-L1 [PD-(L)1]

15) Line 741: cGAS

16) Line 754: Bcl-xL

17) Line 781: ZEB1 and CDH2

18) Line 784: TGF-β

2. In cases where abbreviations are used within figures, please list these abbreviations along with their corresponding full names in the figure legends. If there are two or more abbreviations, arrange them in alphabetical order.

1) Figure 1: OATP and MDRI

2) Figure 2: TIL

3) Figure 3: Topo and TME

4) Figure 4 : HDAC

5) Figure 5: DRP, ABC transporter, EMT, GSH, and HDAC

3. Line 669: The authors, in the midst of discussing carcinogenesis, abruptly introduced content related to apigenin. Does this align with the overall context of the paper? If necessary, it would be appropriate to place reference 240 after the sentence describing apigenin.

4. Figures 1 through 5 only have titles without accompanying explanations of the content. It would be beneficial to enhance reader understanding by adding descriptions to the figure contents and providing full names for all abbreviations used in the respective figure legends. When listing abbreviations, arrange them alphabetically for clarity.

[Minor concerns]

1. Lines 42 and 43: The font size does not match the format of the manuscript.

2. Reduce the spaces immediately preceding the words mentioned below by removing unnecessary single-letter spacing.

1) Line 146: The unique

2) Line 220: Often

3) Line 443: studies

4) Line 451: double

5) Line 454: combinations

6) Line 536: In

7) Line 575: Drug

8) Line 733: to

9) Line 839: The

3. Lines 318, 370, 650, 760, and 776: References must be listed in single parentheses.

4. Line 510: There should be a period at the end of the reference.

5. Line 190: Taxus brevifolia, the Pacific yew or western yew, is a species of tree in the yew family Taxaceae native to the Pacific Northwest of North America. Therefore, ‘Pacific yew Taxus brevifolia’ should be corrected to ‘Taxus brevifolia, Pacific yew,’ and.......

6. Lines 245, 247, 255, 256, and 257: ‘TOP1’ ‘hTOP2A’ ‘hTOP2B’ ‘hTOP2’ should be used as a uniform abbreviation.

7. Lines 268 and 296: The C in ‘Comptothecin’ should be lowercase.

8. Lines 388 and 399: H2O2 should be written as H2O2.

9. Reference section: Author should consult and peruse carefully recent issues of the journal, Cancers, for format and style. Author must modify all the reference section.

Journal Articles:1. Author 1, A.B.; Author 2, C.D. Title of the article. Abbreviated Journal Name Year, Volume, page range.

Overall, the manuscript can be considered to publication after major revision as indicated above.

Author Response

Reviewer 1 Comments:

General Comment: This paper is a highly detailed review of topoisomerase inhibitors as anticancer agents. While overall well-written, there are a few additional points that need to be addressed as highlighted in the following comments.

Response: The authors appreciate important suggestions.

[Major concerns]

Comment: 1. Abbreviations: The use of abbreviations when writing a paper has many advantages besides simplicity of expression. To use an abbreviation, first write the abbreviation in parentheses after the full name, and then use the abbreviation from Introduction to the final Conclusion. Only in Abstract and Figure legend do it separately. In addition, when using abbreviations, use the same abbreviation (Line 41 topo-active vs line 245 TOP, line 118 Topoisomerase II (Top II) vs Line 153, 162 and 163 Top2). If an abbreviation is not used more than twice, there is no need to define it, so please delete it. In the case of an abstract, the use of abbreviations is completely separate from the main text. As you know, there are cases where only the abstract is introduced separately, so in the abstract, abbreviations should only be used if they are repeatedly used and if they are not used again, only the full name should be used.

Response: We have corrected these and now have included a separate page listing abbreviations.

Comment: * There is no definition of abbreviation. Please define the followings.

1) Line 17: ABC transporters

2) Line 19, 22, 25, 41, 46, 54, 63…: Topo-active drugs

3) Line 41, 46, 54, 63…: Topo-active drugs

4) Lines 51 and 52: Topo I and Topo II’

5) Line 67: ROS

6) Line 207: MDR1

7) Line 290: DOX

8) Line 293: NADPH

9) Line 324: BBB

10) Line 381: CPT

11) Line 394: SOD and CAT

12) Line 444: PARP and ATM

13) Line 458: Chk2

14) Line 559: PD-1/PD-L1 [PD-(L)1]

15) Line 741: cGAS

16) Line 754: Bcl-xL

17) Line 781: ZEB1 and CDH2

18) Line 784: TGF-β

Response: The manuscript is modified to include appropriate abbreviations.

Comment: 2. In cases where abbreviations are used within figures, please list these abbreviations along with their corresponding full names in the figure legends. If there are two or more abbreviations, arrange them in alphabetical order.

1) Figure 1: OATP and MDRI

2) Figure 2: TIL

3) Figure 3: Topo and TME

4) Figure 4 : HDAC

5) Figure 5: DRP, ABC transporter, EMT, GSH, and HDAC

Response: A separate abbreviation section is now included. Also, abbreviations and corresponding full names are included in the figure legends

Comment 3. Line 669: The authors, in the midst of discussing carcinogenesis, abruptly introduced content related to apigenin. Does this align with the overall context of the paper? If necessary, it would be appropriate to place reference 240 after the sentence describing apigenin.

Response: The authors agree with comment. We have removed this paragraph for a better clarity.

Comment 4. Figures 1 through 5 only have titles without accompanying explanations of the content. It would be beneficial to enhance reader understanding by adding descriptions to the figure contents and providing full names for all abbreviations used in the respective figure legends. When listing abbreviations, arrange them alphabetically for clarity.

Response: We have added full names of abbreviations.

[Minor concerns]

Comment 1. Lines 42 and 43: The font size does not match the format of the manuscript.

Response: The texts are corrected

Comment 2. Reduce the spaces immediately preceding the words mentioned below by removing unnecessary single letter spacing.

1) Line 146: The unique

2) Line 220: Often

3) Line 443: studies

4) Line 451: double

5) Line 454: combinations

6) Line 536: In

7) Line 575: Drug

8) Line 733: to

9) Line 839: The

Response: Corrected.

Comment 3. Lines 318, 370, 650, 760, and 776: References must be listed in single parentheses.

Response: Corrected

Comment 4. Line 510: There should be a period at the end of the reference.

Response: Corrected

Comment 5. Line 190: Taxus brevifolia, the Pacific yew or western yew, is a species of tree in the yew family Taxaceae native to the Pacific Northwest of North America. Therefore, ‘Pacific yew Taxus brevifolia’ should be corrected to ‘Taxus brevifolia, Pacific yew,’ and.......

Response: Corrected

Comment 6. Lines 245, 247, 255, 256, and 257: ‘TOP1’ ‘hTOP2A’ ‘hTOP2B’ ‘hTOP2’ should be used as a uniform abbreviation.

Response: We have adopted a uniformity for Topoisomerase I (TOPI), Topoisomerase II (TOPII) etc. as suggested.

Comment 7. Lines 268 and 296: The C in ‘Comptothecin’ should be lowercase.

Response: Corrected

Comment 8. Lines 388 and 399: H2O2 should be written as H2O2.

Response: Corrected

Comment 9. Reference section: Author should consult and peruse carefully recent issues of the journal, Cancers, for format and style. Author must modify all the reference section.

Journal Articles:1. Author 1, A.B.; Author 2, C.D. Title of the article. Abbreviated Journal Name Year, Volume, page range.

Response: We have corrected references.

Reviewer 2 Report

Comments and Suggestions for Authors

The manuscript from Nilesh Kumar Sharma et al. is a review dealing with resistance phenomena related to topoisomerase-active drugs. There are already many published reviews dealing with this issue as, for instance, https://doi.org/10.3390/ijms24087233, and I am worried that I do not see any novelty or improvement in the present one. Moreover, the manuscript title is: ”Cracking the Code: Understanding Cancer's Defense against Topoisomerase-Active Drugs: A Comprehensive Review”. Apart from too many punctuations in the title, which must be changed, a reader would expect to know specifically the mechanism of topoisomerase inhibitors/poisons resistance/defense by cancer(s). Instead, the most of the reported mechanisms of resistance described are not specific for topoisomerases (topos) inhibitors/poisons, but can be applied to general mechanisms of acquired cancer resistance. Thus, the title is quite misleading. I do not have “cracked the code” at the end of the reading, actually I felt more confused than before. This trend can be seen all along the whole manuscript (Paragraphs 4, 6, 7, 8) in which general mechanisms of cancer acquired drug resistance have been reported, as well for drugs targeting topos, but not specifically. In more details:

par 4 related to the Topoisomerases poisons/inhibitors. The described mechanisms are general and can be ascribed to other drugs different from those targeting Topos.

par 6. The same descriptions can be done for other chemotherapeutics that do not target Topos.

par 7 describes the role of ABC transporters in the resistance to Topos inhibitors. What is the difference amongst them and the resistance to tamoxifen, for instance, or another anticancer compound? It is a specific mechanism? If so, authors should indicate what it is.

Par 8, GSH depletion. Another general mechanism…I would like to cite a paper published in “Cancers (Basel)”: “GSH biochemistry deregulation in tumors has been observed in many different murine and human cancers. In addition to the properties mentioned above, particularly in cancer cells, GSH is important in the protection against tumor microenvironment-related aggression, apoptosis evasion, colonizing ability, and multidrug and radiation resistance. Increased levels of GSH and resistance to chemotherapeutic agents have been observed, e.g., for platinum containing compounds, alkylating agents (such as melphalan), anthracyclines, doxorubicin, and arsenic”. (10.3390/cancers3011285).

Overall, the authors should edit the title and add some missing important concepts about the topos drugs resistance, potentiating the paragraphs describing the specific mechanisms underlying the topos inhibitors resistance (topos overexpression, mutations, gene regulation, post-transcriptional modifications and so on) and resuming the general mechanisms.

Minor issues

1)       Ellipticine an alkaloid derived from the Ochrosia elliptica labil has been previously approved as a treatment for the metastatic disease of breast cancer. Additionally, several carbazole derivatives from the species have demonstrated inhibitory activity against Topo I and Topo II [61,68]. However, the references 61, 65, 66, 67, 68 are not appropriate, for instance ref 68 does not deal with carbazole derivatives as Topos inhibitors, but as GPER modulators. Proper references would be the following: 10.1016/j.ejps.2016.09.039, 10.1002/cmdc.201800546, 10.3390/ph16030353

2)       Several topo-active drugs, e.g., doxorubicin, etoposide, Camptothecin and its related…”please, uniform the drugs names.

3)       Conclusions are very general, not focusing on the title-declared subject. Authors should revise, resume and be more specific on the topic.

Comments on the Quality of English Language

Moderate editing of writings is required.

Author Response

Reviewer 2 Comment:

General Comment: The manuscript from Nilesh Kumar Sharma et al. is a review dealing with resistance phenomena related to topoisomerase-active drugs. There are already many published reviews dealing with this issue as, for instance, https://doi.org/10.3390/ijms24087233, and I am worried that I do not see any novelty or improvement in the present one. Moreover, the manuscript title is: ”Cracking the Code: Understanding Cancer's Defense against Topoisomerase-Active Drugs: A Comprehensive Review”. Apart from too many punctuations in the title, which must be changed, a reader would expect to know specifically the mechanism of topoisomerase inhibitors/poisons resistance/defense by cancer(s). Instead, the most of the reported mechanisms of resistance described are not specific for topoisomerases (topos) inhibitors/poisons, but can be applied to general mechanisms of acquired cancer resistance. Thus, the title is quite misleading. I do not have “cracked the code” at the end of the reading, actually I felt more confused than before. This trend can be seen all along the whole manuscript (Paragraphs 4, 6, 7, 8) in which general mechanisms of cancer acquired drug resistance have been reported, as well for drugs targeting topos, but not specifically. In more details:

Response: The authors appreciate these suggestions.  Manuscript title is modified for a better clarity.

Comment: par 4 related to the Topoisomerases poisons/inhibitors. The described mechanisms are general and can be ascribed to other drugs different from those targeting Topos.

Response: The reviewer is correct, and this is true. However, topo-active drugs (topo I or topo II)-induce DNA damage utilizing very specific enzymes while other drugs may bind to DNA (non-covalent or covalent) directly e.g., cis-platin to cause DNA damage. The inhibition of topoisomerases (I and II) by topo-active drugs to control DNA structure and inhibit transcription is the most important difference between topo-active drugs and non-topo drugs. Also, repair of topo-active drug-mediated DNA damage is different than non-topo active drugs.

Comment: par 6. The same descriptions can be done for other chemotherapeutics that do not target Topos.

Response: True, same comments as above. But not all chemotherapy drugs have same mechanisms of resistance as topo-active drugs. Anti-metabolites are totally different, different mechanisms of cell kill and different mechanisms of resistance. Alkylating agent, e.g., cis-pt or cyclophosphamide have totally different mechanisms of action and resistance.

Comment: par 7 describes the role of ABC transporters in the resistance to Topos inhibitors. What is the difference amongst them and the resistance to tamoxifen, for instance, or another anticancer compound? It is a specific mechanism. If so, authors should indicate what it is.

Response: While it is true that many topo-active drugs are substrates of ABC transporters, like some non-topo-active drugs e.g., taxol. It is also true that many non-cancer chemotherapy drugs are substrates for ABC-transporters. Here, in this review we have described many mechanisms of resistance to this important class of clinically active drugs; one of them is resistance caused by ABC transporters. Furthermore, we have utilized this to knowledge to suggest various combinations of inhibitors and topo-active drugs to overcome chemotherapy resistance.

Incidentally, the key mechanism of resistance for tamoxifen is loss/decreased of expression ER receptors rather than role of P-gp in decreased cellular concentrations.

Comment: Par 8, GSH depletion. Another general mechanism…I would like to cite a paper published in “Cancers (Basel)”: “GSH biochemistry deregulation in tumors has been observed in many different murine and human cancers. In addition to the properties mentioned above, particularly in cancer cells, GSH is important in the protection against tumor microenvironment-related aggression, apoptosis evasion, colonizing ability, and multidrug and radiation resistance. Increased levels of GSH and resistance to chemotherapeutic agents have been observed, e.g., for platinum containing compounds, alkylating agents (such as melphalan), anthracyclines, doxorubicin, and arsenic”. (10.3390/cancers3011285).

Response: The suggested reference is now included.

General Comments: Overall, the authors should edit the title and add some missing important concepts about the topos drugs resistance, potentiating the paragraphs describing the specific mechanisms underlying the topos inhibitors resistance (topos overexpression, mutations, gene regulation, post-transcriptional modifications and so on) and resuming the general mechanisms.

Response: We have modified the title as recommended and removed some redundant paragraphs for a better clarity. Appropriate references are also included.

Minor issues

Comment 1: Ellipticine an alkaloid derived from the Ochrosia elliptica labil has been previously approved as a treatment for the metastatic disease of breast cancer. Additionally, several carbazole derivatives from the species have demonstrated inhibitory activity against Topo I and Topo II [61,68]. However, the references 61, 65, 66, 67, 68 are not appropriate, for instance ref 68 does not deal with carbazole derivatives as Topos inhibitors, but as GPER modulators. Proper references would be the following: 10.1016/j.ejps.2016.09.039, 10.1002/cmdc.201800546, 10.3390/ph16030353

       Response: The authors appreciate important comments. We have incorporated suggested references.

Comment 2. Several topo-active drugs, e.g., doxorubicin, etoposide, Camptothecin and its related…”please, uniform the drugs names.

Response: The authors have corrected names of drugs and have included their abbreviations for uniformity.

Comment 3. Conclusions are very general, not focusing on the title-declared subject. Authors should revise, resume and be more specific on the topic.

Response: The authors have improved the conclusions.

Comment 4. Comments on the Quality of English Language. Moderate editing of writings is required.

Response: The manuscript is now further edited and have now removed some paragraphs for a better clarity.

Reviewer 3 Report

Comments and Suggestions for Authors

Dear Editors,

The review manuscript “Cracking the Code: Understanding Cancer's Defense Against Topoisomerase-active Drugs: A Comprehensive Review” by Nilesh Kumar Sharma, Anjali Bahot, Gopinath Sekar, Mahima Bansode, Kratika Khunteta, Priyanka Vijay Sonar, Ameya Hebale, Vaishnavi Salokhe, Birandra Kumar Sinha is dedicated to the drug resistance problem. The theme is actual but the text is chaotic and needs to be organized. There are a lot of typos and many inaccuracies in the text. It was impossible to check each of the 300 references, but incorrect citations were found, that is described below in the review. I recommend it for major revision, but it needs a very thorough revision, checking of the references and modifications of some parts of the text.

Lines 42-43: the font is different.

Line 49 and further in the text: the authors are very careless about notation. There are several variants of designations of topoisomerase 2, like Top2, TOP2, Topo II, TOPO II; and topoisomerase 1, like topo I, Topo I. It should be uniform.

Line 67: “ROS” should be deciphered.

Line 107: the reference about phosphorylation is needed.

Lines 129-130 “Similar to topotecan, the Topo I poison acts after the enzyme cleaves DNA and prevents ligation.” -  What does it mean “Similar to topotecan”? Why topotecan was underlined from other poisons? Why there is only one poison similar to topotecan? If there is one, it could be written and should be referenced.

Lines 130-131: “The overexpression of Topo I is correlated with an increase in the susceptibility of tumor cells to Topo I toxins.” – What are Topo I toxins? How do they differ from inhibitors (poisons and suppressors)?

Lines 132-133: “Shikonin and other topo I suppressors, on the other hand, block topoisomerase I from interacting with the DNA cleavage site” – The reference is needed.

Lines 137-138: “Topo I, which alters the topographic state of duplex DNA through single strand breaks and relegation, is identified as the molecular target of Camptothecin (CPT) [51].” – Ref. 51 is wrong. This paper explores the causes of resistance. It was shown earlier that CPT inhibits Topo I.

Lines 140-141: “The CPT Inhibits Topo I by blocking the reconnecting stage of the 140 cleavage/relegation reaction. [52].” – The title of this paper is “Chemotherapy with irinotecan (CPT-11), a topoisomerase-I inhibitor, for refractory and relapsed non-Hodgkin's lymphoma”. It is about irinotecan. It is a wrong reference on the mechanism of CPT inhibition.

Lines 153-154: “Topoisomerase II (Top II, Top2) inhibitors have demonstrated strong clinical effectiveness against several human malignancies [47].” – This paper is focused on Top2 functions, it is better to provide another reference about Top2 inhibitors.

Line 166: What is the difference between Topo IIα and Topo IIβ?

Lines 171-176: ciprofloxacin chalcones hybrids (CP hybrids)

What is the value of the chalcones? Why are there in the same paragraph with NO and the clinical agent etoposide?

Line 178: What are “Mutant epigenetic enzymes”? The reference is needed here.

Lines 181-183: Recent research has suggested that unreported mutation sites in the DOT1 domain of DOT1L are linked to lung cancer treatment resistance. Where is the reference and how is it related to topoisomerases?

Part 3 “Mechanisms of Action of Topo-Active Drugs” does not describe mechanisms of action per se, it has only listing the types of inhibitors and the inhibitors.

Lines 219-220: “Studies show aneuploidy patterns in various cancers influence tumor suppressor genes and oncogenes related drug resistance induced by topo active.” – The sentence is not completed.

Lines 220-222: “Often, large chromosomal alterations, seen as macro evolutionary events, can potentially lead to the development of drug resistance to topo active agents [71,88].” – There is nothing about topo active agents in Ref 88.

Lines 239-241: “Using High throughput technologies it was identified that the TNBC have developed diverse histone modifications, resulting in resistance to topo active drug containing chemotherapies [99].” – wrong reference 99.

Lines 232-240: this part of the text is not related to Tumor Heterogeneity and Topo Active Drugs.

Line 288: and or

Where is ABC transporter on Figure 1?

Line 376: “A clear association between GSH concentration and topo- 374 active drug efficacies is shown by the enhanced anticancer effects of doxorubicin-loaded GSH-Nano sponges in the cancer cells that lead to high GSH content [141,142]” – 142 is a wrong reference.

Line 435: TOP3A appears. The authors should give some information about TOP3A and TOP3B above in 2. Topoisomerases.

Line 451: ds DNA?

Line 466: AML?

Line 638: typo “to thwart

Author Response

Reviewer 3 Comments:

General Comment: The review manuscript “Cracking the Code: Understanding Cancer's Defense Against Topoisomerase-active Drugs: A Comprehensive Review” by Nilesh Kumar Sharma, Anjali Bahot, Gopinath Sekar, Mahima Bansode, Kratika Khunteta, Priyanka Vijay Sonar, Ameya Hebale, Vaishnavi Salokhe, Birandra Kumar Sinha is dedicated to the drug resistance problem. The theme is actual but the text is chaotic and needs to be organized. There are a lot of typos and many inaccuracies in the text. It was impossible to check each of the 300 references, but incorrect citations were found, that is described below in the review. I recommend it for major revision, but it needs a very thorough revision, checking of the references and modifications of some parts of the text.

Response: We appreciate important suggestions. We have modified title of the paper (Understanding Cancer's Defense Against Topoisomerase-active Drugs: A Comprehensive Review) for a better clarity. We have removed the part of title “Cracking the code”. We have corrected typos and edited abbreviations as suggested.

Comment: Lines 42-43: the font is different.

Response: We have corrected texts.

Comment: Line 49 and further in the text: the authors are very careless about notation. There are several variants of designations of topoisomerase 2, like Top2, TOP2, Topo II, TOPO II; and topoisomerase 1, like topo I, Topo I. It should be uniform.

Response: The authors have noted suggestions and adopted to a standard abbreviation as TOP1, TOPII etc. throughout the manuscript. 

Comment: Line 67: “ROS” should be deciphered.

Response: We have corrected texts.

Comment: Line 107: the reference about phosphorylation is needed.

Response: References are now included.

Comment: Lines 129-130 “Similar to topotecan, the Topo I poison acts after the enzyme cleaves DNA and prevents ligation.” -  What does it mean “Similar to topotecan”? Why topotecan was underlined from other poisons? Why there is only one poison similar to topotecan? If there is one, it could be written and should be referenced.

Response: The authors have corrected texts

Comment: Lines 130-131: “The overexpression of Topo I is correlated with an increase in the susceptibility of tumor cells to Topo I toxins.” – What are Topo I toxins? How do they differ from inhibitors (poisons and suppressors)?

Response: We have changed the texts to a standard terminology. Topo I toxins are the similar to inhibitors (poisons and suppressors).

Comment: Lines 132-133: “Shikonin and other topo I suppressors, on the other hand, block topoisomerase I from interacting with the DNA cleavage site” – The reference is needed.

Response: Appropriate references are added.

Comment: Lines 137-138: “Topo I, which alters the topographic state of duplex DNA through single strand breaks and relegation, is identified as the molecular target of Camptothecin (CPT) [51].” – Ref. 51 is wrong. This paper explores the causes of resistance. It was shown earlier that CPT inhibits Topo I.

Response: Appropriate references are added.

Comment: Lines 140-141: “The CPT Inhibits Topo I by blocking the reconnecting stage of the 140 cleavage/relegation reaction. [52].” – The title of this paper is “Chemotherapy with irinotecan (CPT-11), a topoisomerase-I inhibitor, for refractory and relapsed non-Hodgkin's lymphoma”. It is about irinotecan. It is a wrong reference on the mechanism of CPT inhibition.

Response: Appropriate references are added.

Lines 153-154: “Topoisomerase II (TopII, Top2) inhibitors have demonstrated strong clinical effectiveness against several human malignancies [47].” – This paper is focused on Top2 functions, it is better to provide another reference about Top2 inhibitors.

Response: Appropriate references are added.

Comment: Lines 171-176: ciprofloxacin chalcones hybrids (CP hybrids)

Response: The paragraph is removed for a better clarity

Comment: What is the value of the chalcones? Why are there in the same paragraph with NO and the clinical agent etoposide?

Response: The authors appreciate suggestion. This paragraph is not appropriate here along with NO. Therefore, we removed this paragraph for a better clarity.

Comment: Line 178: What are “Mutant epigenetic enzymes”? The reference is needed here.

Response: A detailed section on epigenetic enzymes are provided on epigenetic enzymes such as HDAC in the section 13. “Epigenetic Changes and Topo-Active Drugs”. To avoid repetition, we have removed the content of line 178.

Comment: Lines 181-183: Recent research has suggested that unreported mutation sites in the DOT1 domain of DOT1L are linked to lung cancer treatment resistance. Where is the reference and how is it related to topoisomerases?

Response: This has now been corrected.

Comment: Part 3 “Mechanisms of Action of Topo-Active Drugs” does not describe mechanisms of action per se, it has only listing the types of inhibitors and the inhibitors.

Response: Additional changes/modifications are included.

Comment: Lines 219-220: “Studies show aneuploidy patterns in various cancers influence tumor suppressor genes and oncogenes related drug resistance induced by topo active.” – The sentence is not completed.

Response: This has been corrected now.

Comment: Lines 220-222: “Often, large chromosomal alterations, seen as macro evolutionary events, can potentially lead to the development of drug resistance to topo active agents [71,88].” – There is nothing about topo active agents in Ref 88.

Response: The reference is corrected.

Comment: Lines 239-241: “Using High throughput technologies it was identified that the TNBC have developed diverse histone modifications, resulting in resistance to topo active drug containing chemotherapies [99].” – wrong reference 99.

Response: The references are modified.

Comment: Lines 232-240: this part of the text is not related to Tumor Heterogeneity and Topo Active Drugs.

Response: The text is corrected.

Comment: Line 288: and or

Response: The text is corrected.

Comment: Where is ABC transporter on Figure 1?

Response: Authors appreciate comment. In the figure, Pgp is labelled. Pgp is a part of the class of ABC transporter in the Figure legend.

Comment: Line 376: “A clear association between GSH concentration and topo- 374 active drug efficacies is shown by the enhanced anticancer effects of doxorubicin-loaded GSH-Nano sponges in the cancer cells that lead to high GSH content [141,142]” – 142 is a wrong reference.

Response: References are corrected.

Comment: Line 435: TOP3A appears. The authors should give some information about TOP3A and TOP3B above in 2. Topoisomerases.

Response: The text about TOP3A is included in the line 108 and 109.

Line 451: ds DNA?

Response: Expanded form of ds (double strand) DNA is included.

Comment: Line 466: AML?

Response: Expanded form of AML is included.

Comment: Line 638: typo “to thwart”

Response: This has been corrected.

Round 2

Reviewer 1 Report

Comments and Suggestions for Authors

Accept in present form

Reviewer 2 Report

Comments and Suggestions for Authors

I do not see any relevant improvement with respect to the last revision round, nor the references have been corrected/implemented. Authors replies to the comments have not been included in the v2 version, for instance:

Comment: Par 8, GSH depletion. Another general mechanism…I would like to cite a paper published in “Cancers (Basel)”: “GSH biochemistry deregulation in tumors has been observed in many different murine and human cancers. In addition to the properties mentioned above, particularly in cancer cells, GSH is important in the protection against tumor microenvironment-related aggression, apoptosis evasion, colonizing ability, and multidrug and radiation resistance. Increased levels of GSH and resistance to chemotherapeutic agents have been observed, e.g., for platinum containing compounds, alkylating agents (such as melphalan), anthracyclines, doxorubicin, and arsenic”. (10.3390/cancers3011285).

Response: The suggested reference is now included.

this is not true.

Again:

Comment 1: Ellipticine an alkaloid derived from the Ochrosia elliptica labil has been previously approved as a treatment for the metastatic disease of breast cancer. Additionally, several carbazole derivatives from the species have demonstrated inhibitory activity against Topo I and Topo II [61,68]. However, the references 61, 65, 66, 67, 68 are not appropriate, for instance ref 68 does not deal with carbazole derivatives as Topos inhibitors, but as GPER modulators. Proper references would be the following: 10.1016/j.ejps.2016.09.039, 10.1002/cmdc.201800546, 10.3390/ph16030353

       Response: The authors appreciate important comments. We have incorporated suggested references.

where? again in the reference 68 there are not experiments dealing with topo inhibition

Comments on the Quality of English Language

Writing is sufficient.

Reviewer 3 Report

Comments and Suggestions for Authors

Dear Editors,

The review manuscript “Understanding Cancer's Defense Against Topoisomerase-active Drugs: A Comprehensive Review” by Nilesh Kumar Sharma, et al, still have many inaccuracies in the text. As I wrote before, it was impossible to check each of the 300 references, but I found incorrect citations.

Unfortunately, the authors worked only on the references that were indicated in my review, and even not carefully. Authors should check each reference in a review. This round some more mistakes were found, but it is an authors’ task to make a correct citation especially in the review. The text also still needs very thorough revision.

Here are some examples, but these are probably not all mistakes.

Among the refs 41-47 works 44, 45 are wrong citation.

There is no reference still on TOPI phosphorylation (line 108): “TOPI, a 100 kDa monomeric protein, is expressed by a single copy gene on chromosome 20q12-13.2 which must be phosphorylated for its catalytic activity”.

There is nothing about shikonin in the ref. 51 (Pommier, et al., Repair of topoisomerase I-mediated DNA damage. Prog Nucleic Acid Res Mol Biol. 2006).

 Refs 46 and 47 are about TOPII, but not TOPI as in the text.

“Using High-throughput technologies it was identified that the TNBC has developed diverse epigenetic modifications, resulting in tumor heterogeneity and resistance to topo-active drug-containing chemotherapies [99].” – Authors changed the reference, but there is no fact in this text that the TNBC has developed diverse epigenetic modifications, resulting in resistance to topo-active drug-containing chemotherapies.

Reference [66] (Chen, et al., Am. J. Physiol. Cell Physiol. 2015) is a wrong citation in lines 175 and 754.